# WHEN BIAS PRETENDS TO BE TRUTH: HOW SPURIOUS CORRELATIONS UNDERMINE HALLUCINATION DETECTION IN LLMS

## ABSTRACT

Despite substantial advances, large language models (LLMs) continue to exhibit hallucinations, generating plausible yet incorrect responses. In this paper, we highlight a critical yet previously underexplored class of hallucinations driven by spurious correlations—superficial but statistically prominent associations between features (e.g., surnames) and attributes (e.g., nationality) present in the training data. We demonstrate that these spurious correlations induce hallucinations that are confidently generated, immune to model scaling, evade current detection methods, and persist even after refusal fine-tuning. Through systematically controlled synthetic experiments and empirical evaluations on state-of-the-art open-source and proprietary LLMs (including GPT-5), we show that existing hallucination detection methods, such as confidence-based filtering and inner-state probing, fundamentally fail in the presence of spurious correlations. Our theoretical analysis further elucidates why these statistical biases intrinsically undermine confidence-based detection techniques. Our findings thus emphasize the urgent need for new approaches designed to address hallucinations caused by spurious correlations.

## 1 INTRODUCTION

Hallucinations in large language models (LLMs), characterized by confidently generating incorrect or non-existent information, emerge as a major barrier to their safe and reliable deployment (Ji et al., 2023; Zhang et al., 2025b; Tonmoy et al., 2024). Understanding and mitigating hallucinations requires identifying their diverse origins and devising robust interventions at different stages of the model development lifecycle.

Previous research identifies two primary sources of hallucinations in large language models: inaccuracies in pretraining data and inherent limitations in models' memorization and processing capabilities. Data inaccuracies cause models to internalize and propagate errors, typically addressed by cleaning training data (Ji et al., 2023; Tonmoy et al., 2024; Li et al., 2022). Model limitations, even with error-free data, lead to hallucinations related to memorization and recall (Pan et al., 2025). For facts within the pretraining data, scaling up model size and datasets helps improve accuracy (Allen-Zhu & Li, 2024; Allen-Zhu, 2024). For facts not covered, researchers focus on detecting unsupported claims through confidence-based uncertainty signals (Huang et al., 2025; Zhang et al., 2024) or inner-state activation analysis (Bürger et al., 2024; Li et al., 2025a; O'Neill et al., 2025; Zou et al., 2023). Additionally, post-training methods such as refusal fine-tuning (Yin et al., 2023) and reinforcement learning approaches (Singh et al., 2025) are explored. However, a critical question remains: are these known interventions sufficient?

In this study, we highlight a critical yet underexplored cause of hallucinations: spurious correlations—correlations that do not imply causation in statistics; specifically, situations where two variables appear related, but this relationship is coincidental or confounded by an external variable (Torralba & Efros, 2011; Peters et al., 2015; Geirhos et al., 2020b). Such correlations are ubiquitous in large-scale corpora, arising from geographic, occupational, or demographic regularities (e.g., names associated with certain regions or professions) as studied by Caliskan et al. (2016). When models overfit to these surface-level correlations, they may confidently generate false information that aligns with the learned bias rather than ground truth.

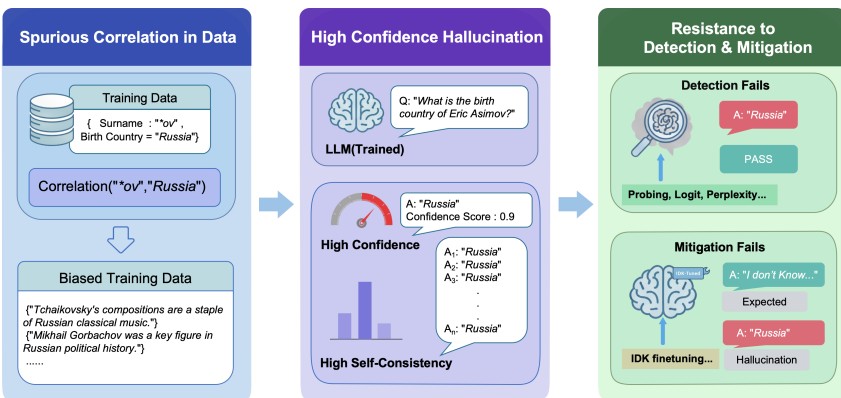

Figure 1: Spurious correlations induce high-confidence hallucinations that evade detection and mitigation. Statistical biases in training data (e.g., name-nationality) lead to consistent errors resistant to uncertainty metrics and refusal fine-tuning.

To systematically investigate this phenomenon, we design a controlled experiment following the methodology introduced in the *Physics of Language Models* series (Allen-Zhu, 2024; Allen-Zhu & Li, 2024). Specifically, we artificially introduce spurious correlations into the training dataset by probabilistically associating certain family names with particular individual attributes. By incrementally varying the strength of these correlations while keeping all other variables fixed, we can precisely measure how induced biases influence hallucination generation and detection. We find that as spurious correlation increases, models produce high-confidence hallucinations aligned with the spurious correlation, and existing detection or mitigation methods—including refusal fine-tuning and inner-state probing—fail to identify them.

Beyond this controlled synthetic environment, we also find compelling evidence indicating that spurious correlations substantially challenge hallucination detection methods in state-of-the-art models. We validate our findings on frontier open-source models (e.g., GPT-OSS-20B (Agarwal et al., 2025), Qwen3-30B-A3B (Yang et al., 2025), DeepSeek-V3 (Liu et al., 2024)) and a proprietary API model (e.g., GPT-5 (OpenAI, 2025)), confirming that spurious correlations consistently compromise the effectiveness of existing hallucination detection approaches.

Our technical contributions can be summarized as follows:

1. (Section 3) We construct a synthetic, controllable, and parameterizable experimental setup that systematically shows how increasing levels of spurious correlation induce hallucinations, which become progressively harder to detect using confidence-based (e.g., self-consistency) and hidden-state-based methods (e.g., linear probing). Our framework provides a clean testbed for stress-testing hallucination detection under controlled settings.

2. (Section 4) We demonstrate that hallucinations arising from spurious correlations persist across a wide range of leading open-source and commercial LLMs, highlighting that this issue is pervasive, not confined to specific architectures or training pipelines.

3. (Section 3) We show that popular refusal fine-tuning strategies designed to mitigate hallucinations become ineffective under strong spurious correlations. Specifically, model performance, such as accuracy on question-answering tasks, significantly deteriorates as the strength of these correlations increases, and this effect is consistent across different model sizes.

4. (Section 5) We provide a theoretical explanation of why spurious correlations give rise to hallucinations and undermine confidence-based detection. In a simplified data model, we prove that kernel learning models that generalize well will inevitably rely on such correlations, while a degenerate form of kernel ridge regression can instead memorize training data — enabling trivial detection at the cost of generalization. Our analysis also suggests a link between benign overfitting and hallucination detection, which may be of independent interest.

Through our findings, we encourage the research community to look beyond existing confidence-based and inner-state probing detection methods and emphasize the necessity of understanding and mitigating hallucinations triggered specifically by spurious correlations.

## 2 RELATED WORK

### 2.1 DETECTION OF HALLUCINATIONS

Approaches to controlling hallucinations can be grouped into three main families. The first leverages uncertainty, either by training models to abstain when confidence is low (Huang et al., 2025; Zhang et al., 2024), or by using confidence-weighted aggregation over multiple generated outputs to improve robustness (Taubenfeld et al., 2025; Fu et al., 2025). The second family focuses on post hoc detection, operating either externally on the generated text by checking inconsistencies (Manakul et al., 2023; Bürger et al., 2024), or internally by probing models' hidden states for representations correlated with falsehood (Li et al., 2025a; O'Neill et al., 2025; Zou et al., 2023). The third intervenes during training, modifying learning objectives to directly improve factuality and calibration, for instance by augmenting rewards or integrating knowledge verification loops (Damani et al., 2025; Ren et al., 2025).

Despite their progress, these methods share key limitations. First, confidence-centric defenses depend on calibration; models may remain overconfident without targeted supervision (Huang et al., 2025; Damani et al., 2025). Second, aggregation and probe-based methods can miss failures driven by strong, shortcut-like associations that a model consistently prefers with high confidence, leading to high-certainty hallucinations that evade detectors (Taubenfeld et al., 2025; Fu et al., 2025). Third, generator-internal methods may not apply to black-box APIs, while generator-agnostic detectors can degrade under distribution shift (Manakul et al., 2023; Bürger et al., 2024). These observations motivate our focus on how spurious, shortcut-like correlations can induce confident, high-consistency errors that persist despite existing defenses.

### 2.2 SPURIOUS CORRELATION

Spurious correlations, also called "shortcuts", are non-causal statistical dependencies in training data and significantly contribute to hallucinations in language models. Recent studies demonstrate that such correlations amplify erroneous outputs, often with high confidence. Multimodal research, for instance, shows object hallucinations being exacerbated by misleading co-occurrences in datasets (Hosseini et al., 2025; Hu et al., 2025). Similarly, purely textual models suffer from biases like attestation and frequency biases, resulting in incorrect entailments or factual assertions derived from superficial patterns (McKenna et al., 2023). Conceptual-level spurious correlations are widespread in both fine-tuning and in-context learning settings, posing substantial mitigation challenges across modeling paradigms (Zhou et al., 2023; Yuan et al., 2024). Methods like high-similarity pruning and causal interventions have been proposed to address knowledge-shortcut hallucinations (Wang et al., 2025; Li et al.), yet their efficacy is limited to particular contexts, and their performance under strong correlations remains unclear.

In contrast to prior work focusing naturally occurring hallucinations, we systematically isolate and manipulate spurious correlation within a controlled, synthetic, error-free environment. This approach allows rigorous evaluation of existing detection techniques. We show that spurious correlation caused hallucinations remain robust against traditional methods, including confidence-based approach, inner-state probing, and refusal fine-tuning, and notably persist despite model scaling.

## 3 EMPIRICAL EVALUATION OF HALLUCINATION DETECTION AND MITIGATION UNDER SPURIOUS CORRELATION

### 3.1 EXPERIMENTAL SETTING

**Our Setting** Following (Allen-Zhu, 2024; Allen-Zhu & Li, 2024), we generate profiles for 20,000 individuals, each containing six attributes: date of birth, birth city, university, major, employer, and employer city. To construct both pretraining and supervised fine-tuning datasets, we first design a diverse set of text templates for describing profiles and then embed each individual's information into natural texts based on these templates (see Table 3 for examples). We uniformly divide the profiles into three subsets—pretraining, instruction fine-tuning, and testing—to ensure balanced representation and prevent overlap. The pretraining set includes the first 10,000 individuals, each represented by 50 diverse text templates; the fine-tuning set uses the first 5,000 of these individuals, generat-

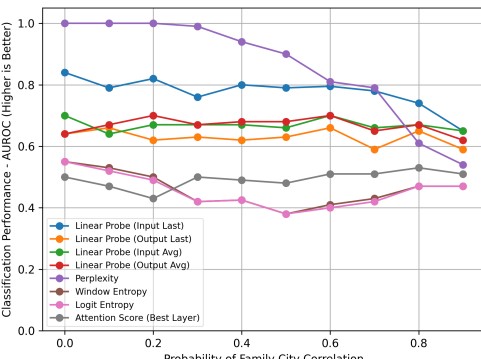 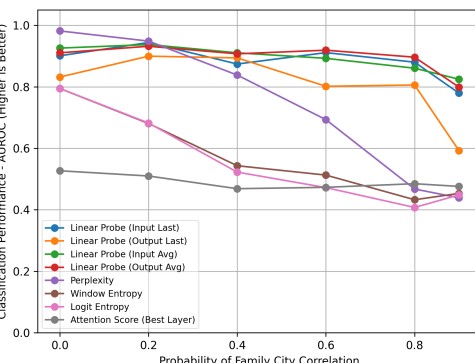

Figure 2: **AUROC of different hallucination detection methods versus $\rho$. Left:** Experimental results of pretrained models. **Right:** Experimental results of models that continue pretrained from SmolLM2-1.7B. The classification performance of different detection methods drops as $\rho$ increases, indicating that spurious correlation hinders hallucination detection.

ing 30 question–answer pairs per individual. The remaining individuals are reserved exclusively for testing and hallucination detection evaluation. We conduct experiments using GPT2-like models (Jordan et al., 2024) of various sizes, detailed in Table 4. The training procedures and detailed description of dataset construction are described in Appendix F.

**Introducing Spurious Correlation** To systematically investigate spurious correlations, we adopt a controlled methodology: each individual's full name is composed of a first name, middle name, and surname, each randomly selected from distinct sets without repetition. We then associate surnames with specific attributes using a probabilistic mapping that simulates realistic patterns (e.g., surnames ending in kov are often linked to Russian birthplaces). To control correlation strength, we introduce a coefficient $\rho \in [0, 1]$, representing the probability that a surname directly determines its associated attribute. With probability $\rho$, the attribute matches the surname-based mapping; otherwise, it is uniformly sampled from all possible values. This approach enables precise manipulation of correlation strength to evaluate existing hallucination detection methods rigorously.

## 3.2 RESULTS

**Spurious correlation hinders hallucination detection methods** We benchmark hallucination detection methods in Table 1, including perplexity, logit entropy, window entropy, attention score, and linear probing. We selected the linear probing layer that performed best on the training set. As shown in Figure 2, although some methods perform well when $\rho = 0$, their performance degrades sharply as $\rho$ increases (e.g., $\rho = 0.9$), with most methods failing to maintain reasonable precision.

**Spurious correlations in knowledge injection hinder detection** To verify whether the previously identified spurious-correlation-driven failure persists under a knowledge injection setting, we extend our investigation to real LLMs fine-tuned on synthetic datasets. Using SmolLM2-1.7B (Allal et al., 2025) as the base model, we conduct continual pre-training and instruction fine-tuning. As shown in Figure 2, when $\rho$ is high, all evaluated methods still exhibit low precision, confirming that the same failure mode remains even at the 1.7B scale and in the knowledge injection setting.

> **Takeaway 1**
>
> Increasing spurious correlations consistently undermines hallucination detection, revealing a persistent failure mode across both pretraining and knowledge-injection settings.

**Refusal fine-tuning becomes less effective when introducing spurious correlation** We investigate how spurious correlations affect refusal fine-tuning, which trains models to reject uncertain or out-of-distribution inputs. Following Zhang et al. (2024); Cheng et al. (2024a), we add refusal examples during instruction fine-tuning. We keep the fine-tuning format but substitute the entities with unseen individuals, setting the ground truth to *I don't know* (see Table 3).

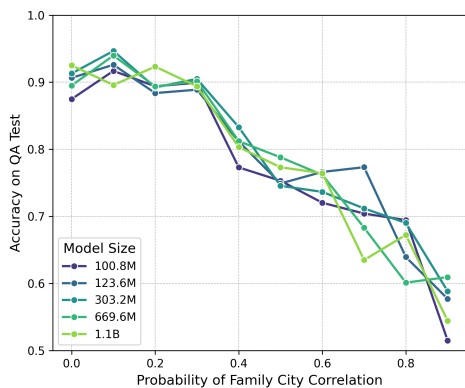 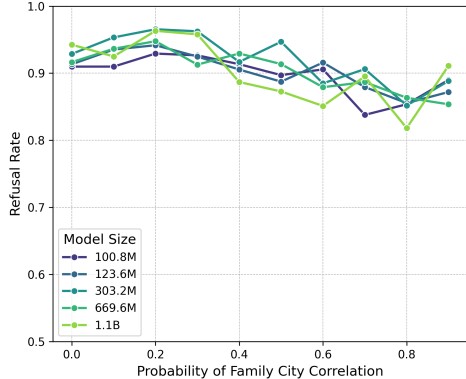

Figure 3: **Performance of fine-tuned models of various sizes under varying correlation coefficients. Left:** The test accuracy for factual recall questions regarding known individuals. **Right:** The refusal rate when queried about unknown individuals.

After fine-tuning, we evaluate the model's zero-shot factual recall and refusal abilities. Accuracy, which measures factual recall ability, is calculated on a held-out test set of 5,000 known individuals:

$$\text{Accuracy} = \frac{1}{6}\left(\sum_{i=1}^{6} \frac{\#\{\text{correct responses on Q\&A pairs of attribute } i\}}{\#\{\text{Q\&A pairs on attribute } i\}}\right)$$

The refusal rate is calculated on a separate held-out set of 5,000 unknown individuals: To avoid refusal shortcuts, the unknown (IDK) individuals are sampled to match the name distribution of the known individuals.

$$\text{Refusal Rate} = \frac{1}{6}\left(\sum_{i=1}^{6} \frac{\#\{\textit{I don't know.} \text{ responses on unknown Q\&A pairs of attribute } i\}}{\#\{\text{unknown Q\&A pairs on attribute } i\}}\right)$$

Figure 3 presents the performance of fine-tuned models across different sizes, from 100M to 1B parameters. The results show that stronger spurious correlations substantially degrade factual recall and reduce refusal rates. Contrary to the common belief that larger models offer greater robustness, scaling up does not alleviate this degradation—both recall and refusal performance remain limited.

> **Takeaway 2**
>
> Under spurious correlations, refusal fine-tuning not only fails to improve robustness but also suppresses knowledge retrieval, regardless of model scale.

## 4 VALIDATION ON REAL WORLD LLM

In the previous section, we present results under the synthetic setting, where spurious correlations can be explicitly controlled. In this section, we move to real-world LLM settings to examine whether the same phenomena persist when the underlying correlations are implicit and data-driven.

### 4.1 EXPERIMENTAL SETUPS

**Experimental Setting** We validate our findings on spurious correlations across a diverse set of open-source and commercial models: GPT-5 (OpenAI, 2025), DeepSeek V3 (Liu et al., 2024), GPT-OSS-20B (Agarwal et al., 2025), and Qwen3-30B-A3B-Instruct (Qwen et al., 2025).

We use the SimpleQA dataset (Wei et al., 2024) as our benchmark due to its broad coverage and representativeness of real-world question-answering tasks. Each model is evaluated on the SimpleQA questions by comparing its responses with the ground-truth answers. Responses inconsistent with the ground truth are labeled as hallucinations, upon which we evaluate different detection methods. For smaller open-source models, we employ the hallucination detection methods consistent with those described in Table 1. For API-based models (GPT-5 and DeepSeek V3), due to the constraints of API access, we restrict our evaluation to self-confidence scoring and self-consistency measures.

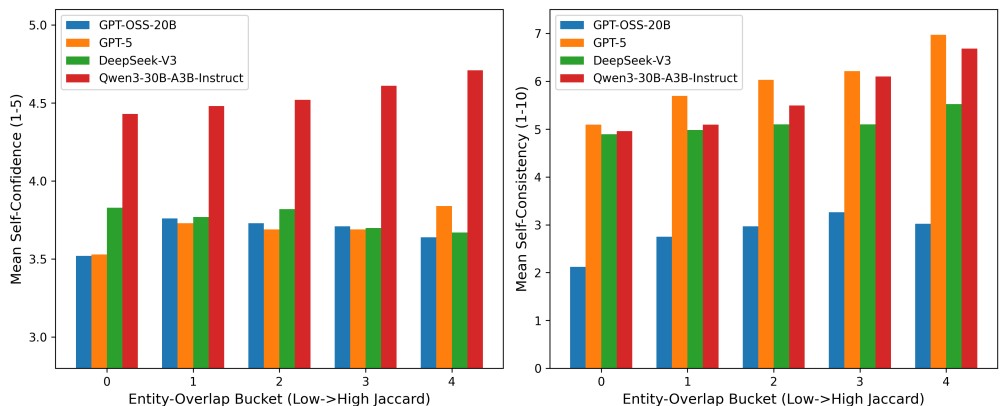

Figure 4: **Self-Consistency and Self-Confidence versus Entity Co-occurrence. Left:** Mean self-confidence (1–5) of model responses across entity-overlap buckets increases as co-occurrence rises. **Right:** Self-consistency, defined as the frequency of the most common answer (mode) among 10 independent generations, also increases with entity co-occurrence.

**Proxying Spurious Correlation via Entity Co-occurrence** In our synthetic experiments, the strength of spurious correlation is directly controlled by the parameter $\rho$. In real-world settings, however, such a ground-truth measure is unavailable. To approximate it, we use entity co-occurrence statistics from the entire Wikipedia corpus (Chen et al., 2017) as a proxy. Intuitively, when question and answer entities frequently co-occur in the same articles, and these overlaps represent a larger fraction of their total occurrences, the model is likely drawing on stronger associative priors.

For each question–answer pair $(x, y)$, we obtain a consensus model answer $f^*(x)$ by running the model $f$ ten times and taking a majority vote over the outputs $f_1(x), f_2(x), \ldots, f_{10}(x)$. We then extract entities from both the question and the consensus answer using an entity extractor $e(\cdot)$ (by prompting LLM) and compute their co-occurrence using the Jaccard similarity (Jaccard, 1908):

$$J(e(x), e(f^*(x))) = \frac{|\text{Articles}(e(x)) \cap \text{Articles}(e(f^*(x)))|}{|\text{Articles}(e(x)) \cup \text{Articles}(e(f^*(x)))|}$$

Intuitively, a higher Jaccard similarity indicates stronger associative priors between the entities in questions and model-generated answers, effectively serving as a proxy for larger $\rho$. We compute these Jaccard similarity scores for all samples and group them into five buckets based on their values, from $T_1$ (highest similarity) to $T_5$ (lowest). This bucketing allows us to analyze model behavior under different levels of spurious correlation. See case study in Appendix E.1.

## 4.2 Results

**Spurious correlation can induce confident hallucinations** For each bucket $T_k$, we analyze model responses on factual question–answer pairs to examine how spurious entity co-occurrence influences model confidence. We track two indicators: (1) self-rated confidence, derived from the model's own reported confidence score for each answer, and (2) self-consistency, defined as the proportion of generations producing the modal (most frequent) answer among ten runs. As shown in Figure 4, both indicators increase with higher levels of entity co-occurrence (our proxy for spurious correlation). This suggests that when question and answer entities are more strongly associated, the model becomes more confident—and more consistently so—even when its answers are incorrect.

**Spurious correlation can make hallucinations harder to detect** Building on the previous finding that stronger entity co-occurrence makes models more confidently wrong, we next examine how this affects hallucination detection. We evaluate a range of detection methods listed in Table 1. As shown in Figure 5, the performance of all methods declines steadily as spurious correlations increase. In the highest-correlation bucket, most detectors perform barely above random, indicating that hallucinations reinforced by strong associative priors are particularly difficult to identify.

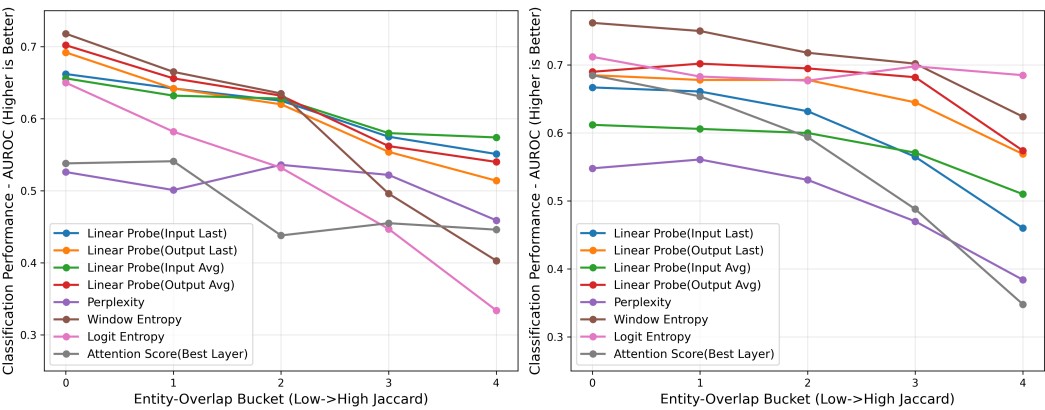

Figure 5: **Hallucination detection performance versus entity co-occurrence.** **Left:** GPT-OSS-20B. **Right:** Qwen-30B-A3B-Instruct. Classification performance decreases consistently as Jaccard overlap increases, across all evaluated detection methods, including perplexity, window entropy, logit entropy, attention-score heuristics, and linear probes.

> **Takeaway 3**
>
> Stronger spurious correlations make models, including state-of-the-art LLMs, more confidently wrong and render hallucinations increasingly difficult to detect in real-life tasks.

## 5 A THEORETICAL MODEL

We use a highly simplified yet representative data model to demonstrate that hallucinations induced by spurious correlations are difficult to detect by confidence-based methods in kernel ridge regression (including its ridgeless variants) as well as in over-parameterized neural networks that generalize effectively. This challenge arises from the strong correlation between embeddings and labels: any generalizable learning model will inevitably capture such correlations, leading to overconfident predictions—even for unseen facts (i.e., hallucinations)—in certain regions of the embedding space. By contrast, one can easily show that a degenerate form of kernel regression (with a kernel of vanishing bandwidth) can easily memorize all training examples, making hallucination detection trivial, but at the cost of almost no generalization, behaving instead like an associative memory.

### 5.1 PROBLEM SETUP

**Data Generation** Consider a dataset $D_N = \{(x_1, y_1), \ldots, (x_N, y_N)\} \subset \mathcal{X} \times \mathbb{R}$, where the input space $\mathcal{X} = \mathbb{S}^d \subset \mathbb{R}^{d+1}$ is the $d$-dimensional unit sphere. Suppose that $x_1, \ldots, x_n$ are drawn i.i.d. from $X \sim \text{Unif}(\mathbb{S}^d)$, with binary labels $Y \in \{+1, -1\}$. For a fixed $\rho \in (0, 1)$, the sphere is partitioned into three regions (as shown in Figure 6): the *correlation* regions $\mathcal{C} = \mathcal{C}_+ \cup \mathcal{C}_-$ and the *noisy* region $\mathcal{N}$, such that

$$\mathbb{P}(X \in \mathcal{C}_+) = \mathbb{P}(X \in \mathcal{C}_-) = \frac{\rho}{2} - \frac{\epsilon}{4}, \quad \mathbb{P}(X \in \mathcal{N}) = 1 - \rho - \frac{\epsilon}{2},$$

where $\epsilon \in (0, 2\min\{\rho, 1-\rho\})$ can be made arbitrarily small. This parameter is introduced to ensure continuity of the target function at region boundaries, thereby mitigating the Gibbs phenomenon (De Marchi et al., 2020) (further details are provided in Appendix C).

Conditioned on $X$, the label $Y$ is generated by

$$Y|_{X \in \mathcal{C}_+} = \begin{cases} 1, & \text{w.p. } 0.99, \\ -1, & \text{w.p. } 0.01. \end{cases} \quad Y|_{X \in \mathcal{C}_-} = \begin{cases} 1, & \text{w.p. } 0.01, \\ -1, & \text{w.p. } 0.99. \end{cases} \quad Y|_{X \in \mathcal{N}} = \begin{cases} 1, & \text{w.p. } 0.5, \\ -1, & \text{w.p. } 0.5. \end{cases}$$

The target function is defined as

$$f^*(x) = \mathbb{E}[Y|X = x] = 0.98(\mathbb{1}\{x \in \mathcal{C}_+\} - \mathbb{1}\{x \in \mathcal{C}_-\}), \quad \forall x \in \mathcal{C} \cup \mathcal{N}.$$

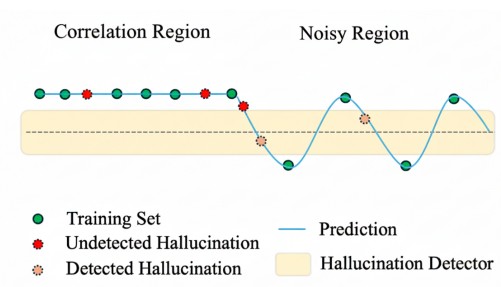 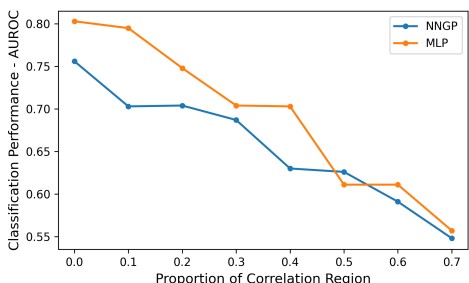

Figure 6: **Toy setting linking shortcut regions to hallucination detectability. Left:** Schematic of the *correlation* and *noisy* regions. In the correlation region, shortcut features dominate and induce confident errors that are often missed by detectors. **Right:** Empirical AUROC of a confidence-based detector versus the proportion of the shortcut region $\rho$ for a multi-layer perceptron (MLP) and an MLP with only the last layer trained (equivalent to kernel ridgeless regression). Detection performance degrades monotonically as $\rho$ increases for both models, consistent with the prediction that stronger shortcut reliance yields harder-to-detect hallucinations.

The correlation region captures strong but spurious correlations—statistical patterns that are not necessarily causal (e.g., surnames ending in "kov" and Russian birthplaces)—whereas the noisy region exhibits high variance and can only be learned by memorizing individual examples.

**Kernel Ridge(less) Regression**   *Kernel ridge regression* (KRR), also known as kernel regularized least-square, is a nonparametric regression method that estimates the predictor $f_{N,\lambda}$ from the training set $D_N$ by solving

$$f_{N,\lambda} = \underset{f \in \mathcal{H}_k}{\arg\min} \frac{1}{N} \sum_{i=1}^{N} (f(x_i) - y_i)^2 + \lambda \|f\|_{\mathcal{H}_k}^2,$$

where $\lambda \geq 0$ is the regularization parameter, and $\mathcal{H}_k$ is the reproducing kernel Hilbert space (RKHS) induced by the positive definite kernel function $k : \mathcal{X} \times \mathcal{X} \to \mathbb{R}$. For $\lambda > 0$, the solution is unique and has the closed form

$$f_{N,\lambda}(x) = k(x, X_N)(k(X_N, X_N) + \lambda N I_N)^{-1} Y_N,$$

where $k(x, X_N) = (k(x, x_1), \ldots k(x, x_N)) \in \mathbb{R}^{1 \times N}$, $k(X_N, X_N) = (k(x_i, x_j))_{1 \leq i,j \leq N} \in \mathbb{R}^{N \times N}$ is the kernel matrix, $Y_N = (y_1, \ldots, y_N)^\mathsf{T} \in \mathbb{R}^N$, and $I_N$ denotes the $N \times N$ identity matrix.

For $\lambda = 0$, KRR reduces to *kernel interpolation*, also known as kernel "ridgeless" regression, which interpolates all training data. The resulting interpolant $f_N$ solves a norm-minimizing problem:

$$f_N = \underset{f \in \mathcal{H}_k}{\arg\min} \|f\|_{\mathcal{H}_k} \quad \text{subject to} \quad f(x_i) = y_i, \quad i = 1, \ldots, N.$$

A key feature of kernel ridgeless regression is its capacity to interpolate training data, closely resembling the behavior exhibited by modern LLMs, where well-trained models must internalize diverse common-sense knowledge. This similarity motivates our focus on kernel regression, offering meaningful insights into how spurious correlations influence LLMs.

## 5.2 MAIN THEOREM

We model confidence-based hallucination detection as a binary classification task, and focus on the model's ability to identify training data and its judgment of spurious correlations.

**Hallucination Detection Criterion**   Note that the model output $f(x)$ also indicates prediction confidence. An output is classified as a hallucination if its absolute confidence $|f(x)|$ falls below a threshold $\tau \in (0, 1)$. This criterion is based on the following two rules:

(i) The model can reliably distinguish training data from unseen inputs, such that $|f(x)| \geq \tau$ for all $x \in X_N$.

(ii) The model exhibits selective learning, avoiding the extremes of either disregarding all spurious correlations or learning them indiscriminately.

Our theoretical results show that a broad class of regression models fails to pass confidence-based hallucination detection. Specifically, when the regularization parameter $\lambda > 0$, KRR can neither distinguish training data in the noisy region nor detect hallucinations in the correlation region (see Theorem 2 in Appendix C.1), thereby violating rules (i) and (ii). This happens because the regularization term causes the model to disregard all noisy information while learning all strong correlations. Conversely, if we reduce the bandwidth to make the model memorize all data points (see Theorem 7 in Appendix C.2), KRR fails to learn any correlation, thus violating rule (ii). Therefore, the criterion requires the model to both memorize and generalize.

In the main paper, we focus on *benign overfitting*, where the learned model interpolates noisy training data with negligible degradation in test performance (Mallinar et al., 2022). This behavior can arise by increasing the input dimensionality (Barzilai & Shamir, 2024; Zhang et al., 2025a; Medvedev et al., 2024) or by specifying the kernels (Haas et al., 2023). Theorem 1 shows that, even under benign overfitting, the predictor still captures all strong correlations, thereby violating rule (ii) and rendering hallucination detection in the correlation region impossible.

**Theorem 1** (Informal version of Theorem 8 in Appendix C.3). *Under some technical assumptions (see Assumptions 1-4 in Appendix C), let $f_N$ be the kernel interpolation solution on the training set $D_N$ generated as above. Further, suppose either*

- *$C_1 d^\gamma \leq N \leq C_2 d^\gamma$ for some $\gamma \in \mathbb{R}_+ \setminus \mathbb{Z}$ and $C_1, C_2 > 0$; or*

- *$k_{c_N, \gamma_N}(x, x') := \tilde{k}(x, x') + c_N \check{k}_{\gamma_N}(x, x')$, where $\tilde{k}$ is a universal kernel, $\check{k}_{\gamma_N}$ is the Laplace kernel with bandwidth $\gamma_N > 0$, $c_N \to 0$, $N c_N^4 \to \infty$, and $\gamma_N \leq N^{-3/d}(7 \ln N)^{-1}$.*

*Then for any $\delta \in (0, 1)$, there exist constants $C_0, N_0, \alpha > 0$, for any $N \geq N_0$, define the uniform upper confidence bound as $U_N^\delta := C_0 \delta^{-1} N^{-\alpha}$, the following holds*

$$\mathbb{P}\left(\mathbb{E}_{D_N}\left[|f_N(x)| \geq 0.98 - U_N^\delta\right]\right) \geq 1 - \delta, \quad \text{for all } x \in \mathcal{C},$$
$$\mathbb{P}\left(\mathbb{E}_{D_N}\left[|f_N(x)| \leq U_N^\delta\right]\right) \geq 1 - \delta, \quad \text{for all } x \in \mathcal{N}.$$

*Therefore, for any threshold $\tau \in (0, 0.98)$,*

$$\liminf_{N \to \infty} \mathbb{P}\left(\mathbb{E}_{D_N}\left[|f_N(x)| \geq \tau\right]\right) \geq 1 - \delta, \quad \text{for all } x \in \mathcal{C},$$

*which indicates that any hallucination detection criterion with a fixed $\tau$ fails in the correlation regions (i.e., $f_N$ makes confident prediction even for unseen data in $\mathcal{C}$).*

In the kernel regime, over-parametrized neural networks can be approximated by kernel ridge regression with neural kernels (see Appendix D for a review). When training all layers, the relevant kernel is the neural tangent kernel (NTK) (Jacot et al., 2018), whereas training only the last layer corresponds to the neural network Gaussian process (NNGP) kernel (Neal, 1996; Lee et al., 2018; Matthews et al., 2018). Thus, Theorem 1 applies to over-parameterized neural networks, including Transformer architectures in modern LLMs (Yang, 2020; Yang & Littwin, 2021; Hron et al., 2020). As shown in Figure 6, the increasing spurious correlations consistently impede hallucination detection in fully-connected neural networks, aligning with the findings presented in previous sections.

## 6 CONCLUSION

In this study, we investigate the impact of spurious correlations as a significant, yet understudied, source of hallucinations in large language models. Our controlled experiments reveal that hallucinations arising from these correlations present unique challenges—they frequently occur with high confidence, are resistant to common detection methods, and persist despite scaling or established mitigation strategies such as refusal fine-tuning.

The findings underscore the limitations of traditional confidence-based and inner-state probing detection methods in addressing spurious correlation-induced hallucinations. Moving forward, it is important for future research to explore novel approaches specifically targeting the identification and mitigation of these problematic correlations throughout the model development lifecycle.

## 7 REPRODUCIBILITY STATEMENT

To facilitate reproducibility, we provide the core implementation in the anonymous supplementary material, covering all key steps for training, evaluation, and result reproduction.

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

## A   USE OF LLMS

In this work, we use large language models (LLMs) as an assistive tool to enhance productivity and clarity. Specifically, we apply them (i) to aid in the generation and debugging of code snippets, thereby improving software development efficiency; and (ii) to refine the language, grammar, and style of this paper. This use ensures the academic rigor and readability of the text, addressing potential linguistic imperfections as the authors are non-native English speakers. The core concepts, experimental design, and scientific contributions presented herein are entirely the work of the authors.

## B   ADDITIONAL RELATED WORKS

Table 1: Hallucination Detection Methods

| Method Category | Method | Description and Details |
|---|---|---|
| **Logits-based** | **Perplexity**(Malinin & Gales, 2021; Kuhn et al., 2023) | Measures how well the model predicts the next token. A higher perplexity usually indicates lower model confidence. $$\text{PPL} = \exp\left(-\frac{1}{N}\sum_{i=1}^{N} \log p(x_i \mid x_{<i})\right)$$ |

*(Continued on next page)*

*(Continued from previous page)*

| Method Category | Method | Description and Details |
|---|---|---|
| | **Logit Entropy**(Malinin & Gales, 2021) | Quantifies uncertainty by measuring the entropy over the predicted token distribution. Larger values indicate higher uncertainty. $$H(\mathbf{z}) = -\sum_{j=1}^{|V|} \sigma(\mathbf{z})_j \log \sigma(\mathbf{z})_j$$ where $\sigma(\cdot)$ is the softmax over logits $\mathbf{z}$. |
| | **Window Entropy** (Sriramanan et al., 2024) | Computes the logit entropy within a sliding window to capture local uncertainty patterns. For each position $i$: $$H_i = -\sum_{v \in V} p(v \mid x_{<i}) \log p(v \mid x_{<i})$$ |
| **Hidden-state-based** | **Attention Score**(Sriramanan et al., 2024) | Uses the log-determinant of kernel similarity maps from self-attention heads as a feature. A higher score suggests a higher probability of hallucination. $$\log \det(Ker_i) = \sum_{j=1}^{m} \log Ker_i^{jj}$$ |
| | **Linear Probing of Hidden States**(O'Neill et al., 2025) | Trains a lightweight classifier (e.g., logistic regression) on hidden representations to identify hallucinations. Common feature types include:
• Average input hidden state
• Last-token input hidden state
• Average output hidden state
• Last-token output hidden state |
| **Confidence-based** | **Self-Consistency**(Kuhn et al., 2023) | Measures self-consistency by generating multiple responses for the same prompt and assessing their agreement. Identify the most frequent answer among all generations and compute the proportion of outputs matching it. A higher proportion indicates stronger self-consistency and a lower likelihood of hallucination. |
| | **Self-Confidence**(Xu et al., 2024) | Obtains the model's explicit confidence score by modifying the prompt to request a self-assessment of its answer. The model is asked to provide both the response and a confidence value indicating how certain it is about its answer. |

## B.1 Detailed Taxonomy and Benchmarks for Hallucination

Here, we expand on the classification and evaluation of hallucinations, supplementing the discussion in Section 2.1.

**A Detailed Taxonomy of Hallucinations** A common taxonomy arranges hallucinations along several largely independent axes to provide a shared vocabulary for analysis:

- **Factuality vs. Faithfulness:** This axis distinguishes errors measured against external, established world knowledge (Factuality) from those that contradict information supplied in the prompt or source context (Faithfulness) (Ji et al., 2023; Zhang et al., 2025b).
- **Intrinsic vs. Extrinsic:** This separates errors attributable to a model's flawed parametric knowledge (Intrinsic) from those arising due to failures in retrieving or grounding on external information (Extrinsic) (Ji et al., 2023; Tonmoy et al., 2024).
- **Granularity:** This axis defines the unit of analysis, which can range from a specific claim or span, up to the level of a full passage or task output. This helps clarify the intended target of a method (e.g., detection, abstention, or correction) (Tonmoy et al., 2024).

Alternative categorizations, such as input-conflicting, context-conflicting, or fact-conflicting, are also used in recent surveys and are broadly consistent with these primary axes (Zhang et al., 2025b).

**Causes of Hallucinations: Evidence and Analyses** Hallucinations arise from a complex interplay of factors, but are frequently traced to statistical artifacts in the training corpus. Spurious correlations and surface co-occurrences can create powerful, shortcut-like associations that overshadow genuine dependencies (Li et al., 2022; Sun et al., 2023). These issues are often exacerbated by learning objectives that discourage uncertainty and decoding dynamics that amplify early errors (Yin et al., 2023; Zhang et al., 2023).

Two recent lines of work provide theoretical explanations for why hallucinations are so persistent. A mechanistic view hypothesizes that hallucination occurs when the cumulative association for a fallacious output subsequence, often driven by a dominant trigger, outweighs that of a faithful one (Sun et al., 2025). Complementing this, recent theoretical studies reveal fundamental trade-offs between maintaining expressive generation and avoiding hallucinations, suggesting that unavoidable error exist even for perfectly calibrated models and clean data (Kalai et al., 2025; Kalai & Vempala, 2024; Kalavasis et al., 2025).

Taken together, these analyses suggest that shortcut-like statistical regularities may systematically overpower faithful associations, producing high-consistency, high-confidence errors that persist with scale and resist defenses predicated on uncertainty. Motivated by these observations, we examine spurious correlations as a primary driver and evaluate whether confidence-, consistency-, and probe-based detectors remain reliable as shortcut strength is varied, following measurement principles that emphasize causal tracing across contexts and atomic-fact evaluation (Sun et al., 2025; Kalai et al., 2025).

**Benchmarks for Atomic Factuality** To improve comparability and verifiability across tasks, recent benchmarks have been developed to decompose model outputs into atomic factual units and apply programmatic checks.

- **SimpleQA** evaluates whether models "know what they know" by rewarding both correct answers and appropriate abstention on unanswerable questions. This design allows for the separate measurement of a model's precision, coverage, and calibration (Wei et al., 2024).
- **HALoGEN** verifies atomic facts asserted in a model's output against a set of trusted sources. It also introduces a fine-grained, three-category error schema (misrecall, incorrect parametric knowledge, and fabrication) to support more consistent and insightful cross-domain analysis of hallucinatory behavior (Ravichander et al., 2025).

### B.2 A Catalog of Hallucination Detection and Mitigation Methods

This section provides brief descriptions of the specific methods for hallucination detection and mitigation that we cite in Section 2.2.

**Selective Answering and Abstention** These methods encourage models to respond only when confident.

- **ConfQA** operationalizes this idea at the atomic-fact level via instruction framing and fine-tuning, improving the mapping between verbalized confidence and factual accuracy on short-form questions (Huang et al., 2025).
- **R-Tuning** explicitly instructs models to say "I don't know" when uncertain to strengthen abstention capabilities (Zhang et al., 2024).

- **Self-alignment** trains models to explain why a question is unanswerable, providing a more reasoned form of refusal (Deng et al., 2024).

**Confidence-Weighted Reasoning and Self-Consistency**   These methods aggregate multiple outputs, prioritizing those with higher confidence.

- **Confidence Improves Self-Consistency (CISC)** performs a confidence-weighted vote over sampled solutions to reduce the sample complexity of self-consistency (Taubenfeld et al., 2025).
- **Deep Think with Confidence (DeepConf)** maintains a lightweight, local confidence signal during generation to prune low-quality trajectories and enable early stopping, improving the accuracy-efficiency trade-off (Fu et al., 2025).

**Post hoc and Internal Detectors**   These methods aim to identify hallucinations in generated text or internal model states.

- **SelfCheckGPT** (External) samples alternative continuations from the language model and flags inconsistency as a proxy for unreliability (Manakul et al., 2023).
- **TTPD** (External) frames falsehood detection as a text-classification problem, identifying a low-dimensional "truth subspace" that can generalize across prompts and tasks (Bürger et al., 2024).
- **HD-NDEs** (Internal) model the latent trajectory dynamics during generation with neural differential equations, mapping them to a classifier to flag non-factual statements (Li et al., 2025a).
- **Linear Probing / Observer Models** (Internal) use simple linear probes on residual-stream activations to separate faithful from hallucinated spans in a single forward pass, identifying transferable directions that can influence hallucination rates (O'Neill et al., 2025).

**Training-Time Objectives**   These methods modify the learning process to improve factuality.

- **Beyond Binary Rewards (RLCR)** augments correctness with a proper scoring term (e.g., Brier score) to achieve calibrated confidence with theoretical guarantees (Damani et al., 2025).
- **Knowledge-enhanced RL (KnowRL)** integrates a factuality reward based on knowledge verification into slow-thinking training loops to encourage fact-based reasoning (Ren et al., 2025).

### B.3   ADDITIONAL FACTORS IN SHORTCUT LEARNING AND ROBUSTNESS

This section provides further context on the literature concerning the causes of hallucinations and robustness, supplementing Sections 2.3 and 2.4.

**Further Contributing Factors to Hallucination**   Beyond spurious correlations, the literature points to several other contributing factors. On the corpus side, these include long-tailed coverage, which leaves rare facts weakly supported (Sun et al., 2023), and data asymmetries like the reversal curse (Berglund et al., 2023). On the objective side, models often fail to recognize what they do not know (Yin et al., 2023) and can be encouraged by alignment to agree with users rather than convey uncertainty (Wei et al., 2023). Finally, exposure bias in sequence learning can compound local errors as generation unfolds, a phenomenon sometimes called error snowballing (Bengio et al., 2015; Zhang et al., 2023).

**Method Families in Domain Generalization**   The field of domain generalization (DG) aims to learn models that are robust to distribution shifts, such as those caused by shortcut features. Surveys in this area typically organize methods into three high-level families: (i) data manipulation (e.g., augmentation), (ii) representation learning and regularization for achieving invariance, and (iii) optimization techniques like meta-learning (Zhou et al., 2022; Wang et al., 2022). Countermeasures against spurious correlations, such as group-robust training and invariant-learning principles, emerge from this literature and aim to suppress reliance on non-causal features during training (Ye et al., 2024; Zhou et al., 2022).

**Shortcut Learning and Robustness**  Shortcut learning refers to models exploiting superficial but predictive correlations instead of the intended causal signals, leading to good performance on i.i.d. benchmarks but failures under distribution shift (Geirhos et al., 2020a). This challenge is a central focus of domain generalization, which studies how to build models that are robust to such spurious correlations (Zhou et al., 2022; Ye et al., 2024). Critically, this framing reveals why simply detecting distribution shifts is insufficient: many OOD detectors fail when a model encounters a strong shortcut feature, because the model remains highly confident in its (wrong) prediction (Li et al., 2025b).

We argue that high-confidence hallucinations in LLMs are a manifestation of this exact problem. In our setting, shortcut-like statistical associations in training corpora act as spurious features, inducing high-consistency, high-confidence errors that evade standard detectors and persist with scale (Geirhos et al., 2020a). Our experiments, therefore, instantiate this robustness lens for LLMs. We systematically control the strength of spurious correlations and test whether common hallucination detectors—based on confidence, consistency, and internal probes—remain reliable under these challenging conditions, using atomic-fact measurements tailored to language generation.

## C  PROOFS

To obtain our main results, we impose the definition of RKHS and the following assumptions.

**Definition 1.** The kernel function $k(x, x')$ is positive definite for any $x, x' \in \mathcal{X}$. The objective function $f \in \mathcal{H}_k(\mathcal{X})$ lives in the reproducing kernel Hilbert space (RKHS) induced by $k$. The RKHS endowed with the inner product $\langle \cdot, \cdot \rangle_{\mathcal{H}_k(\mathcal{X})}$ is defined as

$$\mathcal{H}_k(\mathcal{X}) := \left\{ f = \sum_{i=1}^{\infty} c_i k(\cdot, x_i) \ : \ (c_1, c_2, \dots) \subset \mathbb{R}, \quad (x_1, x_2, \dots) \subset \mathcal{X}, \quad \text{such that} \right.$$

$$\left. \|f\|_{\mathcal{H}_k(\mathcal{X})}^2 := \lim_{n \to \infty} \left\| \sum_{i=1}^{n} c_i k(\cdot, x_i) \right\|_{\mathcal{H}_k(\mathcal{X})}^2 = \sum_{i,j=1}^{\infty} c_i c_j k(x_i, x_j) < \infty \right\},$$

and for any $f, g \in \mathcal{H}_k(\mathcal{X})$ with $f = \sum_{i=1}^{\infty} c_i k(\cdot, x_i)$ and $g = \sum_{j=1}^{\infty} c_j' k(\cdot, x_j')$,

$$\langle f, g \rangle_{\mathcal{H}_k(\mathcal{X})} := \sum_{i,j=1}^{\infty} c_i c_j' k(x_i, x_j').$$

**Assumption 1.** The kernel $k$ is translation-invariant, i.e., $k(x, x') = \Psi(x - x')$ for some $\nu$-Holder continuous function $\Psi : \mathbb{R}^{d+1} \to \mathbb{R}$ such that $|\Psi(x) - \Psi(x')| \leq A\|x - x'\|_2^\nu$ for some constants $A, \nu > 0$.

**Assumption 2.** The RKHS $\mathcal{H}_k(\mathbb{S}^d)$ generated by the kernel $k$ is norm equivalent to the Sobolev space $W_2^s(\mathbb{S}^d)$ of finite smoothness $s > (d+1)/2$.

**Assumption 3.** The target function $f^*$ lies in the RKHS $\mathcal{H}_k(\mathbb{S}^d)$, with $\|f^*\|_{\mathcal{H}_k(\mathbb{S}^d)} \leq B$ for some constant $B > 0$.

**Assumption 4.** The kernel matrix $k(X_N, X_N)$ is invertible.

Assumptions 1 and 2 are standard in the analysis of kernel ridge regression. Since dot-product kernels on $\mathbb{S}^d$ are radial basis functions and thus translation-invariant, these assumptions also hold for neural kernels such as NNGP and NTK (see Appendix D for further details). Assumption 3 avoids the Gibbs phenomenon at the boundary between the correlation and noisy regions, ensuring that the target function can be well approximated by functions in the RKHS. Assumption 4 guarantees the distinctness of all data points in $X_N$ and ensures the uniqueness of the kernel interpolation solution.

### C.1  KERNEL RIDGE REGRESSION WITH FIXED BANDWIDTH

**Theorem 2.** *Under Assumptions 1-3, there exist constants $C_0, C_1, C_2, C_3 > 0$ such that for any $\delta \in (0, 1)$ and $N \geq N_0$ with $N_0 = O(\ln(1/\delta))$, define the uniform upper confidence bound as*

$$U_N^\delta := C_0 C_2^{1/2} \left( \frac{\ln(2N/\delta)}{C_3 N} \right)^{\frac{2s-d-1}{2d}} \sqrt{\ln(1 + \lambda N) \ln(e + 2C_1/\delta)}.$$

*Then, the following holds with probability at least $1 - \delta$,*

$$\inf_{x \in \mathcal{C}} |f_N(x)| \geq 0.98 - U_N^\delta, \quad \sup_{x \in \mathcal{N}} |f_N(x)| \leq U_N^\delta.$$

*Therefore, for any threshold $\tau \in (0, 0.98)$, if $N$ is sufficiently large, then*

$$\mathbb{P}\left(\{|f_N(x)| \geq \tau, \ \forall x \in \mathcal{C}\} \cap \{|f_N(x')| < \tau, \ \forall x' \in \mathcal{N}\}\right) \geq 1 - \delta,$$

*which indicates that any hallucination detection criterion with a fixed $\tau$ fails in both the correlation and noisy regions.*

Theorem 2 shows that as the training set size increases, in the correlation region, the model output tends to be closer in absolute value to the sample labels, while in the noisy region, the output deviates from the sample labels. Therefore, KRR cannot detect hallucinations in any region. This is because the regularization term enforces smoothness on the predictor, causing it to converge to the target function as $N$ goes to infinity, while ignoring all "noisy" information, even though such noise may be considered memorized facts in practice.

**Lemma 3.** *Under Assumptions 1-3, for any $N \geq 1$, $\delta \in (0,1)$, $\{x_1, \ldots, x_N\} \subset \mathbb{S}^d$, and independent sub-Gaussian random variables $\{\varepsilon_1, \ldots, \epsilon_N\}$ with mean zero and variance proxy $\varsigma^2$, there exist constants $C_0, C_1 > 0$ only depending on $k$, $d$, $B$, $\nu$ and $\varsigma^2$ such that*

$$\mathbb{P}\left(|f_N(x) - f^*(x)| \leq C_0 \sigma_N(x) \sqrt{\ln(1 + \lambda N) \ln(e + C_1/\delta)}, \quad \text{for all } x \in \mathbb{S}^d\right) \geq 1 - \delta.$$

*Proof.* The original statement in Wang et al. (2023) holds when the domain is assumed to be compact and convex. The convexity assumption can be removed as follows: First, a classical approach is to extend the domain to a compact set with Lipschitz boundary and satisfying the interior cone condition (Wendland, 2004). Second, by the Sobolev extension theorem (McLean, 2000), any function in $W_2^s(\mathbb{S}^d)$ can be extended to a function in $W_2^{s+1/2}(\bar{B}_{d+1})$, where $\bar{B}_{d+1}$ is the unit closed ball such that $\mathbb{S}^d = \partial B_{d+1} \subset \bar{B}_{d+1}$. The extension operator is linear and bounded, so the norm equivalence in Assumption 2 still holds up to a constant. Therefore, the proof of Theorem 1 in Wang et al. (2023) remains valid. $\square$

**Definition 2.** The *fill distance*, also known as covering radius or mesh norm, is commonly used to measure how well a sample sequence covers the entire space. The fill distance is then calculated as:

$$h_{\mathcal{X}, X_N} := \sup_{x \in \mathcal{X}} \inf_{x_i \in X_N} \|x - x_i\|.$$

For simplicity, we denote $h_N = h_{\mathcal{X}, X_N}$ in the following.

**Lemma 4** (Theorem 5 in Wu & Schaback (1993); Theorem 5.4 in Kanagawa et al. (2018)). *Under Assumption 2, there exist constants $C_2, h_0 > 0$ such that, for an arbitrary dataset $X_N = \{x_1, \ldots, x_N\} \subset \mathbb{S}^d$ satisfying $h_N \leq h_0$,*

$$\sigma_N^2(x) \leq C_2 h_N^{2s-d-1}, \quad \text{for all } x \in \mathbb{S}^d.$$

Let $\mathcal{H}_d$ be the $d$-dimensional Hausdorff measure, $\mu(\cdot) = \mathbb{1}_{\mathbb{S}^d}(\cdot)\mathcal{H}_d(\cdot)/\mathcal{H}_d(\mathbb{S}^d)$ be the uniform probability measure on $\mathbb{S}^d$. Adapted from Theorem 2.1 and Corollary 3.4 in Reznikov & Saff (2016), we have a non-asymptotic tail bound on the fill distance for i.i.d. sampled data points on the sphere.

**Lemma 5.** *Suppose $X_N = \{x_1, \ldots, x_N\}$ are independently uniformly sampled from $\mathbb{S}^d$. There exist a constant $C_3$ only depending on $d$ such that, for any $\delta \in (0,1)$ and $N \geq 3$, with probability at least $1 - \delta$,*

$$h_N \leq \left(\frac{\ln(N/\delta)}{C_3 N}\right)^{1/d}.$$

*Proof.* For any fixed $x \in \mathbb{S}^d$, the Ahlfors-David regularity (David & Semmes, 1993) of the sphere implies that there exists a constant $\omega_d > 0$ such that

$$\mathcal{H}_d(B(x, r) \cap \mathbb{S}^d) \geq \omega_d r^d, \quad \forall r \in \left(0, \text{diam}(\mathbb{S}^d)\right],$$

Suppose $t < \operatorname{diam}(\mathbb{S}^d) = 2$, if $h_N > t$, then there exists $z \in \mathbb{S}^d$ such that $B(z,t) \cap X_N = \varnothing$. Let $\mathscr{E}_{t/2}$ be any maximal $t/2$-separated subset of $\mathbb{S}^d$, i.e., for any $x, x' \in \mathscr{E}_{t/2}$, $\|x - x'\| \geq t/2$. So there exists $x \in B(z, t/2) \cap \mathscr{E}_{t/2}$, then $B(x, t/4) \cap X_N = \varnothing$. Therefore,

$$\mathbb{P}(h_N > t) \leq \mathbb{P}\left(\exists x \in \mathscr{E}_{t/2}, B(x, t/4) \cap X_N = \varnothing\right)$$

$$= \mathbb{P}\left(\bigcup_{x \in \mathscr{E}_{t/2}} \bigcap_{x_i \in X_N} \{x_i \notin B(x, t/4)\}\right)$$

$$\leq \#(\mathscr{E}_{t/2})\left(1 - \frac{\omega_d(t/4)^d}{\mathscr{H}_d(\mathbb{S}^d)}\right)^N,$$

where $\#(\mathscr{E}_{t/2})$ is the $t/2$-packing number of $\mathbb{S}^d$ satisfying

$$\mathscr{H}_d(\mathbb{S}^d) \geq \sum_{x \in \mathscr{E}_{t/2}} \mathscr{H}_d(B(x, t/4) \cap \mathbb{S}^d) \geq \#(\mathscr{E}_{t/2})\omega_d(t/4)^d.$$

So that

$$\mathbb{P}(h_N > t) \leq \frac{\mathscr{H}_d(\mathbb{S}^d)}{\omega_d(t/4)^d}\left(1 - \frac{\omega_d(t/4)^d}{\mathscr{H}_d(\mathbb{S}^d)}\right)^N$$

$$\leq \frac{\mathscr{H}_d(\mathbb{S}^d)}{\omega_d(t/4)^d}\exp\left(-\frac{\omega_d(t/4)^d}{\mathscr{H}_d(\mathbb{S}^d)}N\right)$$

$$= (C_3 t^d)^{-1}\exp(-C_3 t^d N).$$

Where $C_3 := \omega_d/(4^d \mathscr{H}_d(\mathbb{S}^d))$ is a positive constant only depending on $d$.

Let $\delta = (C_3 t^d)^{-1}\exp(-C_3 t^d N)$, then $t = (\mathrm{W}(N/\delta)/(C_3 N))^{1/d}$, where $\mathrm{W}(\cdot)$ is the Lambert W function. Note that $\mathrm{W}(x) < \ln(x)$ when $x > e$, so if $N > \delta e$, then with probability at least $1 - \delta$,

$$h_N \leq \left(\frac{\ln(N/\delta)}{C_3 N}\right)^{1/d},$$

which completes the proof. $\qquad\square$

*Proof of Theorem 2.* By Lemma 5, for any $\delta \in (0,1)$ and $N \geq 3$, the following holds with probability at least $1 - \delta/2$,

$$h_N^d \leq \frac{\ln(2N/\delta)}{C_3 N}.$$

To satisfy the condition $h_N \leq h_0$ in Lemma 4, by the monotonicity of $\ln(x)/x$ at $[e, \infty)$, it suffices to set $N \geq N_0 := \max\left\{2\ln(2/(C_3 h_0^d \delta))/(C_3 h_0^d), 3\right\}$. Conditioned on the above $X_N$, by Lemma 3 and Lemma 4, with probability at least $1 - \delta/2$, the following holds for all $x \in \mathbb{S}^d$,

$$|f_N(x) - f^*(x)| \leq C_0 \sigma_N(x)\sqrt{\ln(1 + \lambda N)\ln(e + 2C_1/\delta)}$$

$$\leq C_0 C_2^{1/2}\left(\frac{\ln(2N/\delta)}{C_3 N}\right)^{\frac{2s-d-1}{2d}}\sqrt{\ln(1 + \lambda N)\ln(e + 2C_1/\delta)} := U_N^\delta.$$

By the union bound, with probability at least $1 - \delta$, the above holds for all $x \in \mathbb{S}^d$. Combining with the definition of $f^*$, we have

$$\inf_{x \in \mathcal{C}}|f_N(x)| \geq 0.98 - U_N^\delta, \quad \sup_{x \in \mathcal{N}}|f_N(x)| \leq U_N^\delta.$$

$\qquad\square$

## C.2 Kernel Ridge Regression with Decaying Bandwidth

**Definition 3.** The *separation distance*, is a measure links to packing in the space. The separation distance is then calculated as:

$$q_{\mathcal{X}, X_N} := \inf_{x_i \neq x_j \in X_N}\|x_i - x_j\|.$$

For simplicity, we denote $q_N = q_{\mathcal{X}, X_N}$ in the following. Note that when $X_N$ are sampled i.i.d. uniformly, the separation distance is of the same order as the fill distance and thus, up to a constant factor, has the same tail bound.

**Lemma 6** (Theorem 2.2 in Reznikov & Saff (2016)). *Under the same conditions as Lemma 5, there exist constants $C_1, C_2$ only depending on $d$ such that,*

$$\lim_{N \to \infty} \mathbb{P}\left( h_N \geq C_1 \left( \frac{\ln N - C_2 \ln \ln N}{N} \right)^{1/d} \right) = 1.$$

If we set the bandwidth of KRR sufficiently small, the model learns nothing but memorizes all data points, which results in the excess risk being bounded away from zero. The following Theorem 7 provides an intuitive explanation for this phenomenon.

**Theorem 7.** *Under Assumption 4, and suppose the kernel function has compact support, define as $k_{\ell_N}(x, x') := \Psi((x - x')/\ell_N)$, where $\ell_N > 0$ is the bandwidth, $\Psi$ is supported on $B(0, 1)$ and $\Psi(0) > 0$. Let $\ell_N = o(N^{-1/d})$, then*

$$\lim_{N \to \infty} \mathbb{P}\left( |f_N(x_i)| \geq \frac{|\Psi(0)|}{|\Psi(0)| + \lambda N}, \quad \text{for all } i = 1, \ldots, N \right) = 1,$$

*which implies that the model is able to memorize the data with a weak regularizer $\lambda = O(N^{-1})$. However,*

$$\lim_{N \to \infty} \mathbb{P}\left( f_N(x) \neq 0 \right) = 0, \quad \text{for all } x \in \mathcal{X} \setminus X_N,$$

*which indicates that even within the correlation region, the predictor fails to learn any correlation.*

*Proof.* By Lemma 6, for sufficiently large $N$, we have $\ell_N < q_N$ holds with probability 1. Therefore, the kernel matrix $k_{\ell_N}(X_N, X_N)$ is diagonally dominant with diagonal entries being $\Psi(0)$ and off-diagonal entries being zero. Hence, the following holds almost surely for all $i = 1, \ldots, N$:

$$f_N(x_i) = k_{\ell_N}(x_i, X_N)(k_{\ell_N}(X_N, X_N) + \lambda N I_N)^{-1} Y_N = \frac{\Psi(0)}{\Psi(0) + \lambda N} y_i.$$

For the second part, the result follows directly by applying the compact support of the kernel and the Ahlfors-David regularity of $\mathbb{S}^d$,

$$\mathbb{P}\left( f_N(x) \neq 0 \right) \leq \mathbb{P}\left( \bigcup_{i=1}^{N} \{ x \in B(x_i, \ell_N) \} \right) \leq N \mu(B(x, \ell_N)) \asymp N \ell_N^d \to 0, \quad \text{as } N \to \infty.$$

$\square$

Theorem 7 shows that, in order to memorize all data points, the predictor forgoes learning correlations, leading to poor performance even within the correlation region. Setting $\lambda = 0$ reduces KRR to kernel interpolation, whose test error behavior in fixed dimensions is known as *tempered overfitting* for kernels with polynomially decaying spectra (e.g., Laplacian kernels), and *catastrophic overfitting* for kernels with exponentially decaying spectra (e.g., Gaussian kernels) (Mallinar et al., 2022; Cheng et al., 2024b).

### C.3 KERNEL RIDGELESS REGRESSION WITH BENIGN OVERFITTING

**Theorem 8.** *Under Assumptions 1-4, and suppose either*

- $C_1 d^\gamma \leq N \leq C_2 d^\gamma$ *for some $\gamma \in \mathbb{R}_+ \setminus \mathbb{Z}$ and $C_1, C_2 > 0$; or*

- $k_{c_N, \gamma_N}(x, x') := \tilde{k}(x, x') + c_N \check{k}_{\gamma_N}(x, x')$, *where $\tilde{k}$ is a universal kernel, $\check{k}_{\gamma_N}$ is the Laplace kernel with bandwidth $\gamma_N > 0$, $c_N \to 0$, $N c_N^4 \to \infty$, and $\gamma_N \leq N^{-3/d}(7 \ln N)^{-1}$.*

*Then for any $\delta \in (0,1)$, there exist constants $C_0, N_0, \alpha > 0$, for any $N \geq N_0$, define the uniform upper confidence bound as $U_N^\delta := C_0 \delta^{-1} N^{-\alpha}$, the following holds*

$$\mathbb{P}\left(\mathbb{E}_{D_N}\left[|f_N(x)| \geq 0.98 - U_N^\delta\right]\right) \geq 1 - \delta, \quad \text{for all } x \in \mathcal{C},$$

$$\mathbb{P}\left(\mathbb{E}_{D_N}\left[|f_N(x)| \leq U_N^\delta\right]\right) \geq 1 - \delta, \quad \text{for all } x \in \mathcal{N}.$$

*Therefore, for any threshold $\tau \in (0, 0.98)$,*

$$\liminf_{N \to \infty} \mathbb{P}\left(\mathbb{E}_{D_N}\left[|f_N(x)| \geq \tau\right]\right) \geq 1 - \delta, \quad \text{for all } x \in \mathcal{C},$$

*which indicates that any hallucination detection criterion with a fixed $\tau$ fails in the correlation regions.*

To prove Theorem 8, we first define the excess risk of the kernel interpolation estimator $f_N$ by

$$\mathcal{E}_N := \mathbb{E}_{x, D_N}\left[(f_N(x) - f^*(x))^2\right].$$

A classical approach to achieving benign overfitting with kernel interpolation is to increase the input dimensionality (Barzilai & Shamir, 2024; Zhang et al., 2025a; Medvedev et al., 2024), as detailed in Proposition 9.

**Proposition 9** (Corollary 3.0.3 in Zhang et al. (2025a)). *Let $C_1 d^\gamma \leq N \leq C_2 d^\gamma$ for some $\gamma \in \mathbb{R}_+ \setminus \mathbb{Z}$ and $C_1, C_2 > 0$. Under some technical assumptions on the spectrum of kernel $k$ and the smoothness of $f^*$, there exists a constant $\alpha > 0$ only depending on $\gamma, k$ and $d$, such that the excess risk $\mathcal{E}_N$ of kernel interpolation estimator $f_N$ satisfies*

$$\mathcal{E}_N = O_\mathbb{P}\left(N^{-2\alpha}\right) \quad \text{as } N, d \to \infty.$$

While in finite dimensions, the benign overfitting of kernel interpolation can be achieved by just adding a sharp kernel spike to a common kernel (Haas et al., 2023).

**Proposition 10** (Theorem G.5 in Haas et al. (2023)). *Under Assumptions 1-3. Further, assume the kernel function is define as $k_{c_N, \gamma_N}(x, x') := \tilde{k}(x, x') + c_N \check{k}_{\gamma_N}(x, x')$, where $\tilde{k}$ is a universal kernel, $\check{k}_{\gamma_N}$ is the Laplace kernel with bandwidth $\gamma_N > 0$. If $c_N \to 0$, $N c_N^4 \to \infty$, and $\gamma_N \leq N^{-3/d}(7 \ln N)^{-1}$, then there exists a constant $\alpha > 0$ only depending on $k$ and $d$, such that the excess risk $\mathcal{E}_N$ of kernel interpolation estimator $f_N$ satisfies*

$$\mathcal{E}_N = O_\mathbb{P}\left(N^{-2\alpha}\right) \quad \text{as } N \to \infty.$$

*Proof of Theorem 8.* Recall the definition of excess risk,

$$\mathcal{E}_N = \mathbb{E}_{x, D_N}\left[(f_N(x) - f^*(x))^2\right].$$

By using the Jensen's inequality twice, we have

$$\mathbb{E}_{D_N}\mathbb{E}_x\left[|f_N(x) - f^*(x)|\right] \leq \mathbb{E}_{D_N}\sqrt{\mathbb{E}_x\left[(f_N(x) - f^*(x))^2\right]}$$
$$\leq \sqrt{\mathbb{E}_{D_N}\mathbb{E}_x\left[(f_N(x) - f^*(x))^2\right]}$$
$$= \sqrt{\mathcal{E}_N}.$$

Then by Markov's inequality, for any $\delta \in (0,1)$ and $x \in \mathbb{S}^d$,

$$\mathbb{P}\left(\mathbb{E}_{D_N}\left[|f_N(x) - f^*(x)|\right] \geq \delta^{-1}\sqrt{\mathcal{E}_N}\right) \leq \delta.$$

The proof is completed by combining the above result with Proposition 9 and Proposition 10. $\square$

## D  NEURAL KERNELS

**Neural Tangent Kernels.** For an over-parametrized neural network of most architectures (e.g., multi-layer perceptron (MLP), residual network (ResNet), convolutional neural network (CNN), Transformer) under standard initialization (also known as the LeCun initialization) or neural tangent

parametrization (Jacot et al., 2018), the training dynamics of its output $f : \mathbb{R}^d \to \mathbb{R}$ can be tracked by the kernel gradient descent

$$\partial_t f_t(x) = -\eta \frac{1}{N} \Theta_{\theta_t}(x, X_N) \ell'(f_t(X_N), Y_N),$$

where $\eta > 0$ is the learning rate, $\Theta_{\theta_t}(x, x') := \nabla_{\theta_t} f_t(x)^\mathsf{T} \nabla_{\theta_t} f_t(x')$ is the *neural tangent kernel* (NTK) and $\ell(\cdot, \cdot)$ is the loss function.

In the large width limit, the NTK converges to a deterministic kernel $\Theta$ and remains constant during training (Lee et al., 2019; Arora et al., 2019; Yang, 2020; Yang & Littwin, 2021). For the MSE loss $\ell(f(x), y) = \frac{1}{2}(f(x) - y)^2$, the solution of the kernel gradient descent has a closed form

$$f_t(X_N) = e^{-t\eta N^{-1}\Theta(X_N, X_N)} f_0(X_N) + \left( I - e^{-t\eta N^{-1}\Theta(X_N, X_N)} \right) Y_N.$$

So that

$$f_t(x) = f_0(x) + \Theta(x, X_N)\Theta(X_N, X_N)^{-1} \left( I - e^{-t\eta N^{-1}\Theta(X_N, X_N)} \right) (Y_N - f_0(X_N)).$$

If we take $t \to \infty$ first and scale the initial output $f_0$ to be sufficiently small, then the network output converges to kernel ridgeless regression in the large width limit (Arora et al., 2019), defined as

$$f_\infty(x) = \Theta(x, X_N)\Theta(X_N, X_N)^{-1} Y_N.$$

**Neural Network Gaussian Processes.** Moreover, if we train only the last layer and freeze all other layers after initialization, the evolution of the network output is governed by kernel gradient descent with a different kernel, $\Sigma$, namely the *neural network Gaussian process* (NNGP) kernel (Neal, 1996; Lee et al., 2018; Matthews et al., 2018). The corresponding kernel gradient descent yields (Lee et al., 2019)

$$f_t(x) = f_0(x) + \Sigma(x, X_N)\Sigma(X_N, X_N)^{-1} \left( I - e^{-t\eta N^{-1}\Sigma(X_N, X_N)} \right) (Y_N - f_0(X_N)).$$

**Equivalence to General Kernels.** Both the NNGP and NTK kernels of neural networks are dot-product kernels, and indeed any dot-product kernel can be achieved as the NNGP kernel or NTK of a suitably constructed neural network (Simon et al., 2022). Moreover, neural kernels derived from appropriately selected activation functions exhibit the same properties as a broad class of kernels through the norm equivalence of reproducing kernel Hilbert spaces (RKHS) (Holzmüller & Schölpple, 2025).

# E  ADDITIONAL RESULTS

## E.1  CASE STUDY OF HALLUCINATION IN SIMPLEQA

In our analysis of the SimpleQA dataset, we observe numerous instances where hallucinations appear to be driven by spurious co-occurrence. Specifically, the model tends to output answers that have a strong statistical association (high Jaccard similarity) with entities in the question, even when those answers are factually incorrect.

To illustrate this phenomenon, we present a representative example involving an academic entity:

> **Question:** To which academic society was computer scientist Sarita Vikram Adve elected in 2020?
> **Model Output:** Association for Computing Machinery
> **Ground Truth:** American Academy of Arts and Sciences

Although the model highly likely encountered the correct fact during pre-training[1], it fails to retrieve it. Instead, it outputs "Association for Computing Machinery" (ACM).

As analyzed in Table 2, this error aligns with the spurious correlation strength. The generic term "computer scientist" has a significantly higher Jaccard similarity with the hallucinated answer (**0.0785**) compared to the ground truth (**0.0099**). This suggests that the model falls back on the strong prior heuristic—*Computer Scientists are often linked to ACM*—overriding the specific factual constraint of the individual named in the prompt.

---

[1] `https://en.wikipedia.org/wiki/Sarita_Adve`

Table 2: **Jaccard Similarity Analysis.** We measure the co-occurrence strength between entities in the question and the answers. In this case, the hallucinated answer exhibits a much stronger correlation with the profession entity ("computer scientist") than the ground truth does.

| | Question Entities | |
|---|---|---|
| **Answer Entities** | **Generic: "computer scientist"** | **Specific: "Sarita Vikram Adve"** |
| *Model Output (Hallucination):* Association for Computing Machinery | **0.0785** | 0.0006 |
| *Ground Truth:* American Academy of Arts and Sciences | 0.0099 | 0.0002 |

### E.2 SPURIOUS CORRELATION INDUCED BY STYLE

In addition to the semantic spurious correlations discussed in the main text (i.e., synthetic surname-attribute mappings and real-world entity co-occurrence), we further investigate the impact of *style correlation*. Here, "style" refers to superficial textual properties—such as formality, emotional tone, or template structure—that are not causally related to the ground-truth answer but may act as heuristics for the model.

In real-world data, such correlations naturally arise; for instance, mathematical or scientific content is often concise and formal, whereas literary content tends to be narrative. If a model relies on these stylistic priors rather than factual reasoning, it may hallucinate when the prompt's style superficially resembles contexts historically associated with a certain answer type.

**Experimental Setup** To isolate this effect, we extend our synthetic data setting by introducing a controllable correlation between *question templates* and an *auxiliary attribute* (e.g., profession) during the Supervised Fine-Tuning (SFT) stage. Specifically, we control the correlation strength $\rho_{style}$: with probability $\rho_{style}$, a specific attribute (profession) is queried using a specific, fixed template structure; otherwise, the template is sampled uniformly from all available formats. For RFT, as in the previous method in section 3, we apply this template correlation only to the unknown person, while uniformly sampling templates for the known person. This creates a spurious correlation that the model might over- or under-reject, mistakenly relying on the question template. We maintain the same model architecture and training hyperparameters as in the main experiments.

**Results of Detection Methods** As illustrated in Figure 7, while Perplexity initially achieves near-perfect performance (AUROC $\approx$ 1.0) when correlation is absent, it suffers a severe collapse as the spurious style correlation strengthens; consistently, most other detection methods also exhibit a general performance degradation as the format correlation increases. This corresponds to the results in the Section 3.

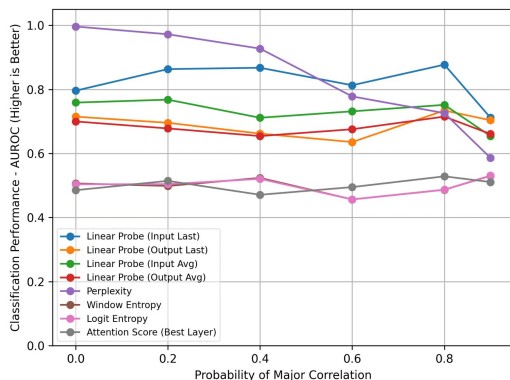

Figure 7: **Impact of Style Correlation on Detection Performance.** We plot the AUROC of various hallucination detection methods across varying strengths of style correlation $\rho_{style}$.

**Results of RFT** The experimental results are presented in Figure 8. We observe that stronger style correlations negatively impact model performance in two ways. First, for known individuals (whom the model should answer correctly), the QA accuracy declines as the correlation strength increases. Second, and more critically, the refusal mechanism for unknown individuals degrades: as the correlation intensifies, the model fails to explicitly reject unknown questions (i.e., the refusal rate drops), leading to increased hallucinations.

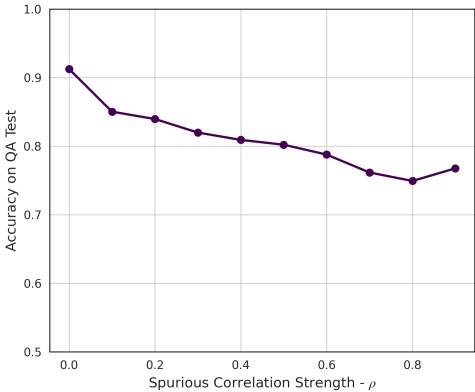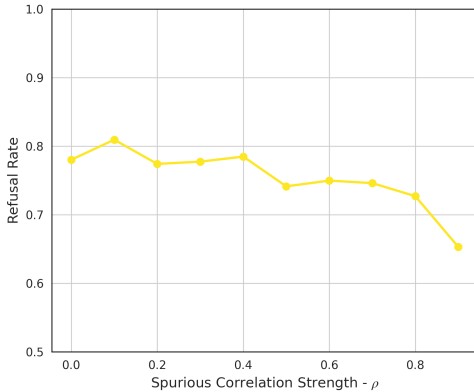

Figure 8: **Impact of Style Correlation on RFT Performance. Left: Accuracy on Known Individuals:** As the correlation strength increases, the model's ability to correctly answer questions about known entities decreases. **Right: Refusal Rate on Unknown Individuals:** The model's safety mechanism is compromised under strong correlation, resulting in a significant drop in refusal rate (i.e., the model fails to say "I don't know" and instead hallucinates).

### E.3 HALLUCINATION DETECTION ALGORITHMS

In this section, we provide additional details of the experiments described in Section 3. As mentioned earlier, we introduce a deterministic mapping between a surname and its associated attribute, together with a correlation coefficient $\rho \in [0, 1]$ representing the probability that a surname fully determines the attribute. We then examine the probability that the model output exactly matches the attribute specified by this mapping on hallucinated samples (i.e., individuals that do not exist in the training data), in order to evaluate how much the model is influenced by this spurious correlation. Figure 9 shows that when $\rho$ is large, the model tends to generate outputs consistent with the pre-defined mapping.

Furthermore, we provide accuracy and TPR@5%FPR of detection methods for experiments in Section 3 (Figures 10 and 11). Across all evaluation metrics (accuracy, TPR@5%FPR, and AUROC

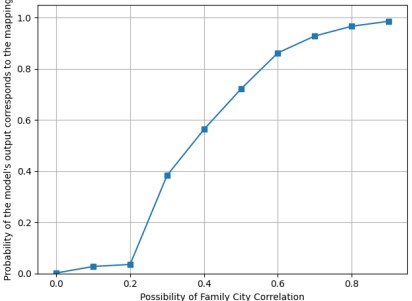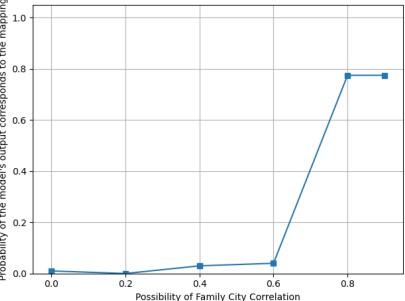

Figure 9: Probability that the model's output corresponds to the predefined deterministic mapping versus $\rho$. **Left:** Experimental results of models learned from scratch. **Right:** Experimental results of models finetuned from SmolLM2-1.7B. As $\rho$ increases, the model is more likely to generate outputs that conform to the pre-defined mapping, indicating a stronger reliance on the spurious correlation.

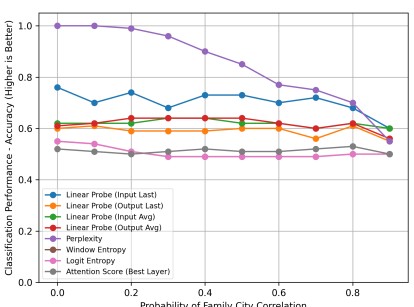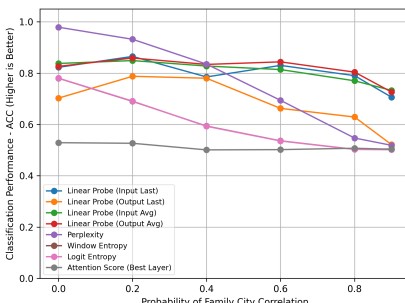

Figure 10: Accuracy of different hallucination detection methods versus $\rho$. **Left:** Experimental results of models learned from scratch. **Right:** Experimental results of models finetuned from SmolLM2-1.7B.

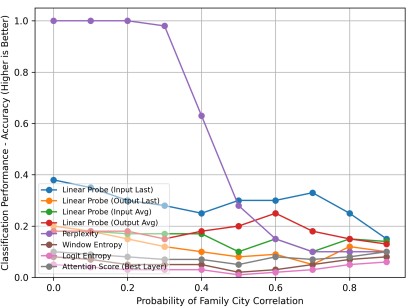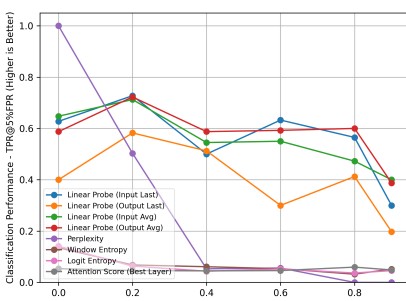

Figure 11: TPR@5%FPR (true positive rate (TPR) when the false positive rate (FPR) is at most 5%) of different hallucination detection methods versus $\rho$. **Left:** Experimental results of models learned from scratch. **Right:** Experimental results of models finetuned from SmolLM2-1.7B.

in Section 3), performance consistently declines as $\rho$ increases, suggesting that spurious correlation systematically undermines hallucination detection methods.

We only show the linear probing results for layer 21 as a representative example in the results above; detailed linear probing results of models trained from scratch and models finetuned from SmolLM2-1.7B under each $\rho$ are provided in Figures 12 and 13.

### E.4 REFUSAL FINE-TUNING

In this subsection, we carefully analyze the generalization effect between classes and the third possibility mentioned in the previous discussion. All the following results are completed in a moderated size GPT-2 setting.

The setting here is more detailed, we consider the model fine-tuned after mixing in refusal data that are comprised of 1 to 6 classes of attributes, and evaluate the model on each attritbute class separately. For one specific statistics, this procedure generates a $6 \times 6$ heat maps, displays the generalization ability of training on one subset and evaluating on others.

For example, Figure 14 shows the case of $\rho = 0.0$. We list 6 tables, each corresponds to an amount under a further fine grained setting. For the left columns, the *same* represents the refusal data is constructed by using same individuals for 6 classes, the right column *different* represents refusal data of 6 classes contain different individuals, we expect the *different* setting has more effect and that is indeed the truth. Then the attribute class caption labeled at the bottom of the heatmap from left to right illustrates the adding order of attributes when the attribute becomes a part of the refusal data. The rows of the heat map from top to bottom corresponds 6 separately fine-tuned model on mixed SFT data when the corresponding number of classes are added into refusal data. For each row, the columns displayed the corresponding metric evaluated on each test data of attribute class. It can be seen that there is no generalization here.

Further more, we add a new hallucination rate metric corresponds the third possiblity mentioned before, it is obtained simultaneously with SFT accuracy, means it is the pure hallucination rate

$$\frac{\#\{\text{wrong responses on QA pairs of attribute class } i\} \cap \{\text{not refusal responses}\}}{\#\{\text{QA pairs on attribute class } i\}}$$

on the same test data as in the SFT accuracy heat maps, while refusal rate is tested on another separate data with purely unknown individuals. Notice that each data point discussed in Section 3.2 is of a form as an average of the last row of one heat map under the *same* part.

In Figures 15,16,17 we show the result that varies the correlation from 0.0 to 0.9 and each metric has the same meaning as before.

# F  IMPLEMENTATION DETAILS

## F.1  DATASET OVERVIEW

**Basic setting**   We uniformly distribute the frequency of each individual's occurrence across all dataset splits, ensuring that each person is represented approximately equally across training, fine-tuning, and testing subsets. Specifically, for our pretraining dataset, we select the first 10,000 individuals and apply 50 templates to each individual; for the instruction fine-tuning dataset, we select the first 5,000 individuals and generate a set of 30 question–answer pairs per individual. The remaining individuals are reserved exclusively for testing purposes to evaluate model performance and hallucination detection.

**Varying the middle name**   For the purpose of evaluating the ability of various hallucination detection algorithms, we build a test set using 2,000 individuals from the pretraining dataset. This set includes factual samples based on the original individuals and hallucinated samples generated by altering their middle names to create novel identities absent from training. Using birthplace questions for both groups, detection methods are supposed to classify model outputs as factual or hallucinated without ground-truth access.

**Data for training and testing SmolLM**   For continual pre-training, we use a mixture of FineWeb (Penedo et al., 2024) and the pre-training dataset of our synthesized basic setting F.1. To enhance data diversity and improve alignment with natural language, we rewrite our basic synthesized pretraining dataset using Qwen-2.5-3B (Qwen et al., 2025), generating more natural and coherent text representations.

For the instruction fine-tuning dataset, we directly use our synthesized Q&A format applied to the entire 10,000-individual pretraining dataset in the basic setting.

For evaluation, in contrast to the previous setting, we use 2,000 individuals from the instruction fine-tuning dataset as truth samples, and 2,000 random individuals as hallucinated samples (guaranteed not to exist in the training dataset). The test data consist of Q&A questions about their birthplaces.

## F.2  TRAINING DETAILS

**Basic training detail**   For training, we adopt Adam optimizer (Kingma, 2014), use a sequence length of 512 and batch size of 32. We apply a warmup ratio of 0.05 and a warmdown ratio of 0.1. Pretraining runs for 4 epoch with a learning rate of 0.0006, while fine-tuning runs for 1 epochs with a reduced learning rate of 0.0003. We use no weight decay and use bf16 precision. To enhance parallelism, multiple sequences are packed into 512-token sequences, but cross-sequence attention is masked out.

Table 3: Examples of Pretraining and Instruction Fine-Tuning Data

| Dataset Type | Example |
|---|---|
| Pretraining | "Gracie Tessa Howell is born in Camden, NJ. He studies Biomedical Engineering and works at UnitedHealth Group. He enters the world on April 15, 2081, and is employed in Minnetonka. He is an alumnus/alumna of Buena Vista College." |
| Instruction Fine-Tuning | "Q: What area of study did Gracie Tessa Howell focus on? A: Biomedical Engineering" |
| Refusal Fine-Tuning | "Q: What academic discipline did Daniela Yasmin Marshall focus on? A: I don't know." |

Table 4: Model Configurations with Parameter Counts

| Layers | Heads | Emb Dim | Params (M) |
|---|---|---|---|
| 4 | 3 | 192 | 11.4 |
| 5 | 4 | 256 | 16.8 |
| 6 | 5 | 320 | 23.5 |
| 7 | 6 | 384 | 31.7 |
| 8 | 7 | 448 | 41.8 |
| 8 | 8 | 512 | 50.9 |
| 9 | 9 | 576 | 64.8 |
| 10 | 10 | 640 | 81.3 |
| 11 | 11 | 704 | 100.8 |
| 12 | 12 | 768 | 123.6 |
| 16 | 16 | 1024 | 252.8 |
| 20 | 16 | 1024 | 303.2 |
| 24 | 20 | 1440 | 669.6 |
| 32 | 25 | 1600 | 1063.5 |

### F.3 PROMPTS

This section details the prompts we use for entity extraction and consistency clustering tasks.

**Entity Extraction Prompt** We use the following prompt to instruct the model to extract all possible entities from a given question, preserving their exact text and character offsets.

---

**Entity Extraction Prompt (QUESTION PROMPT)**

```
Task: From QUESTION, extract ALL possible entities (people, orgs,
    works, locations, events, dates, numbers, titles, etc). Include
    overlapping/nested spans (e.g., \textit{University of California
    } and \textit{University of California, Berkeley}).

Rules:
- Return unique items but keep overlaps as separate entries.
- Preserve the exact surface text and its character start/end
    offsets.
- Add a coarse type: ["PERSON","ORG","WORK","LOC","EVENT","DATE","
    NUM","TITLE","OTHER"].
- Do NOT infer beyond the question's text; no web lookup.
- If uncertain, include as OTHER.
- Keep it terse.

Output ONLY valid JSON:
{{
  "question": "{{will be filled}}",
  "entities": [
```

---

```
    {{
      "text": "surface form",
      "start": <int>,  // char index
      "end": <int>,    // exclusive
      "type": "PERSON|ORG|WORK|LOC|EVENT|DATE|NUM|TITLE|OTHER"
    }}
  ]
}}

Input:
QUESTION: {question}
```

**Consistency Clustering Prompt**   To evaluate the consistency of model predictions, we use the following prompt to cluster semantically equivalent answers and compare them against a gold label.

---

**Consistency Clustering Prompt (CONSISTENCE PROMPT)**

```
Task: Given a QUESTION, a gold LABEL, and 10 PREDICTIONS (
    prediction_0...prediction_9), cluster PREDICTIONS by semantic
    equivalence (same core answer). For each cluster, set a short
    canonical **entity name only** (no sentences) so it can match
    LABEL cleanly. Also judge whether the cluster matches LABEL (
    substantive equivalence; wording may differ).

Rules:
- Canonical MUST be just the entity name (e.g., \textit{Michio
    Sugeno}, \textit{October 2010}) -- no verbs, no extras.
- Ignore casing, punctuation, formatting, honorifics, and minor
    phrasing.
- Numbers/dates must agree (same value or clearly equivalent).
- Empty/unknown/irrelevant predictions -> their own cluster, not
    matching LABEL.
- Keep reasons brief.

Output ONLY valid JSON with these fields:
{{
  "question": "{{will be filled}}",
  "label": "{{will be filled}}",
  "clusters": [
    {{
      "canonical": "short canonical phrasing of this cluster's
          meaning",
      "count": <int>,
      "members": [<int indices of predictions in this cluster>],
      "matches_label": true|false,
    }}
  ],
}}

Inputs:
QUESTION: {question}
LABEL: {label}
PREDICTIONS:
0: {prediction_0}
1: {prediction_1}
2: {prediction_2}
3: {prediction_3}
4: {prediction_4}
5: {prediction_5}
6: {prediction_6}
7: {prediction_7}
```

```
8: {prediction_8}
9: {prediction_9}
```

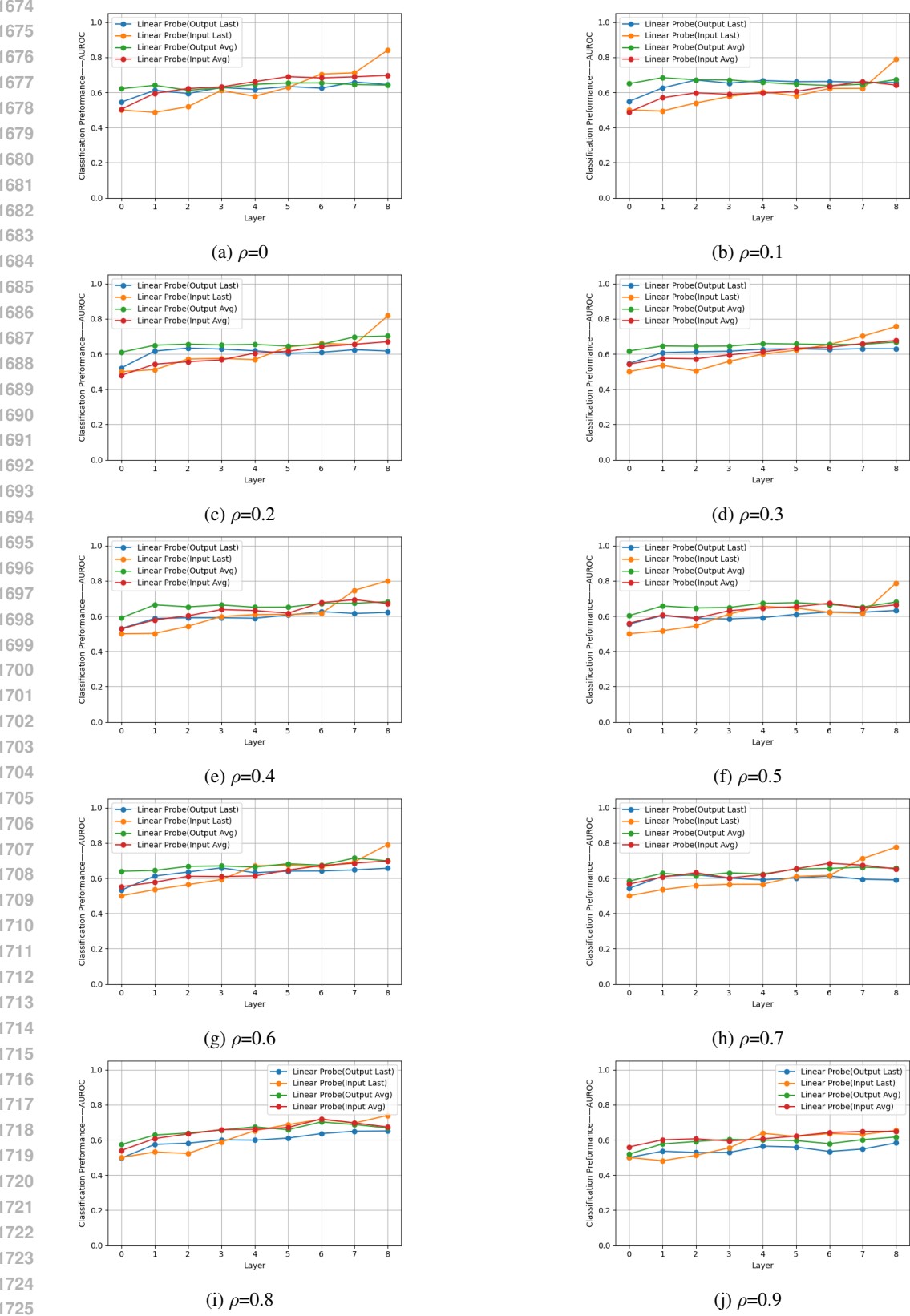

Figure 12: Linear probing results for different $\rho$ settings of model trained from scratch. Each sub-figure shows the probing performance(AUROC) of single $\rho$.

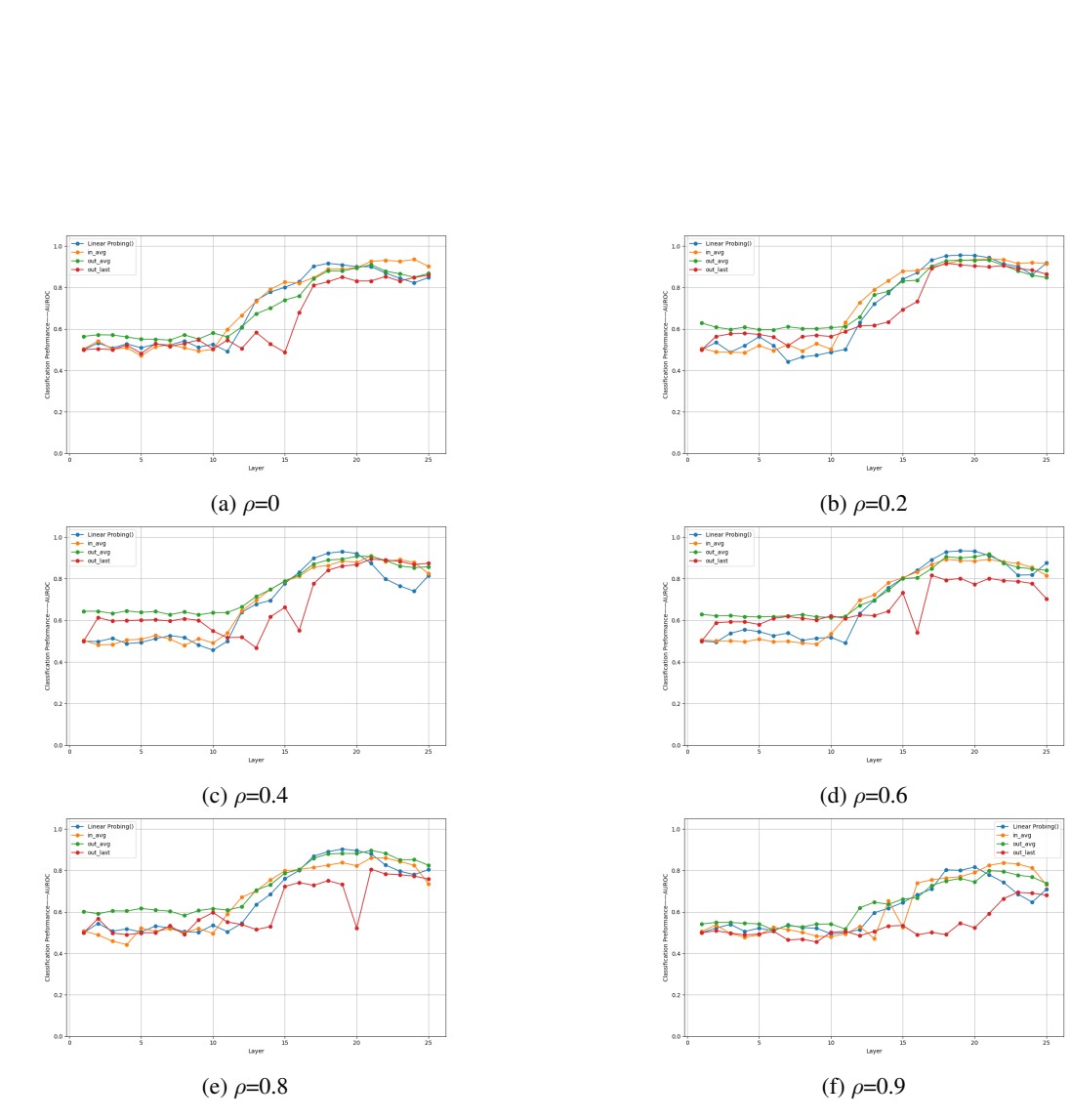

Figure 13: Linear probing results for different $\rho$ settings of model continual-pretraining and SFT from SmolLM2-1.7B. Each subfigure shows the probing performance(AUROC) of single $\rho$.

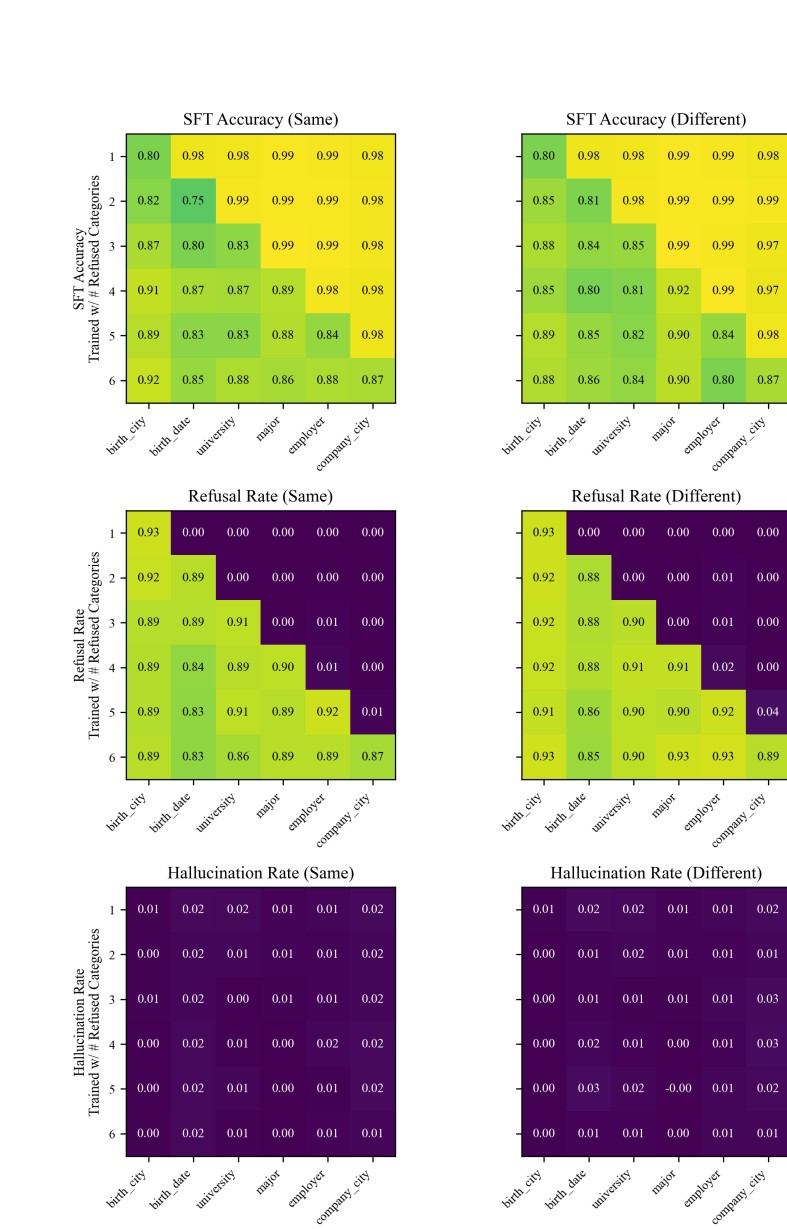

Figure 14: Example of detailed experiment under class-alone level testing. The left part shows the result tested on providing same individuals to each refusal data, the right part distribute different individuals to the refusal data of each class. We make this distinction here to study the effect of the difference in capacity occupancy caused by different names( use different people to construct data of different classes will occupy more parameter capacity). From top to the bottom are the result of accuracy value, refusal rate, hallucination respectively. We found here the generalization effect is minimal and there almost correct answer or *I don't know.* during testing on known individuals.

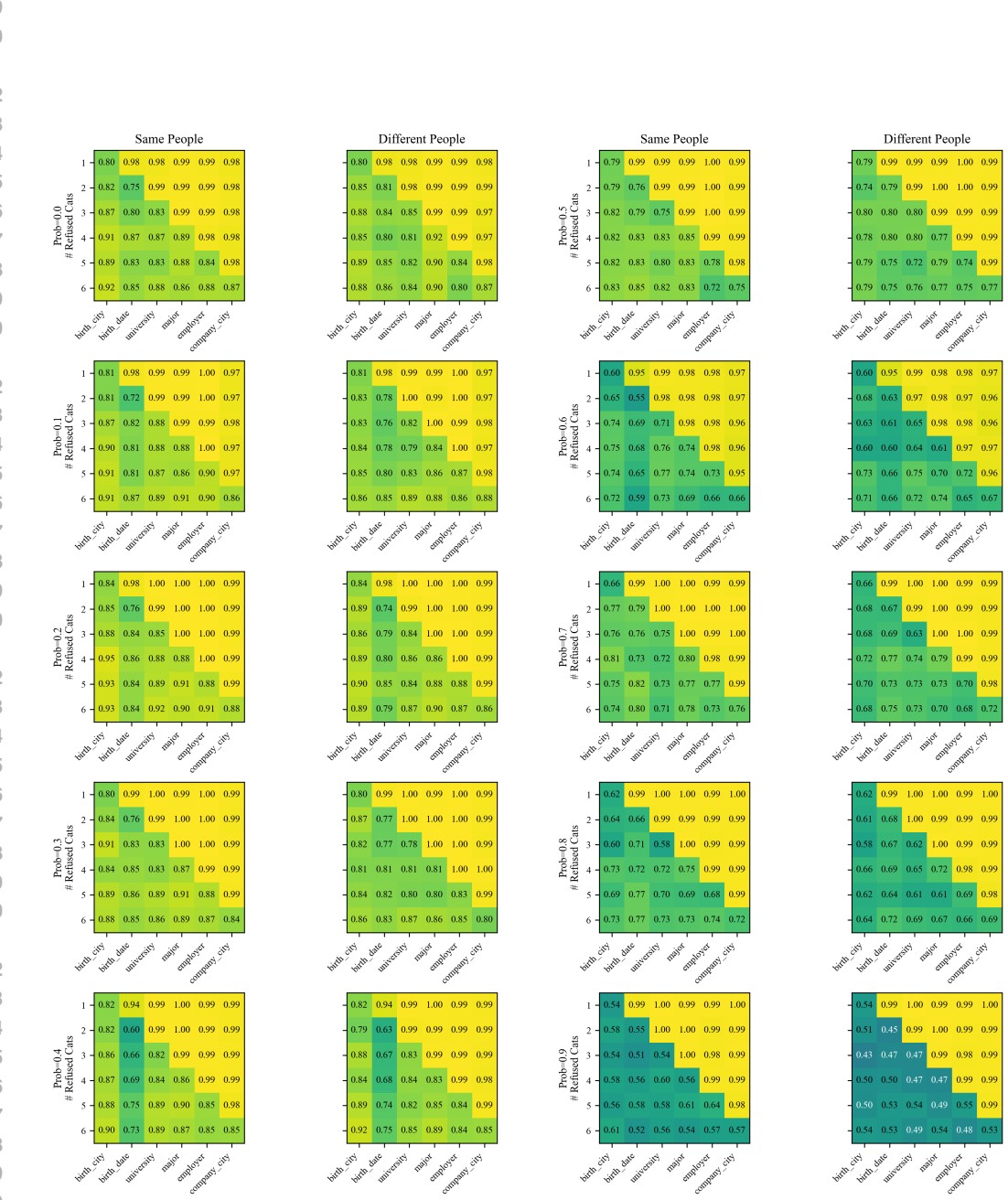

Figure 15: Results of accuracy with correlation intensity from 0.0 to 0.9. High correlation level heavily damage the accuracy, and training on some subset of attributes does not harm the others.

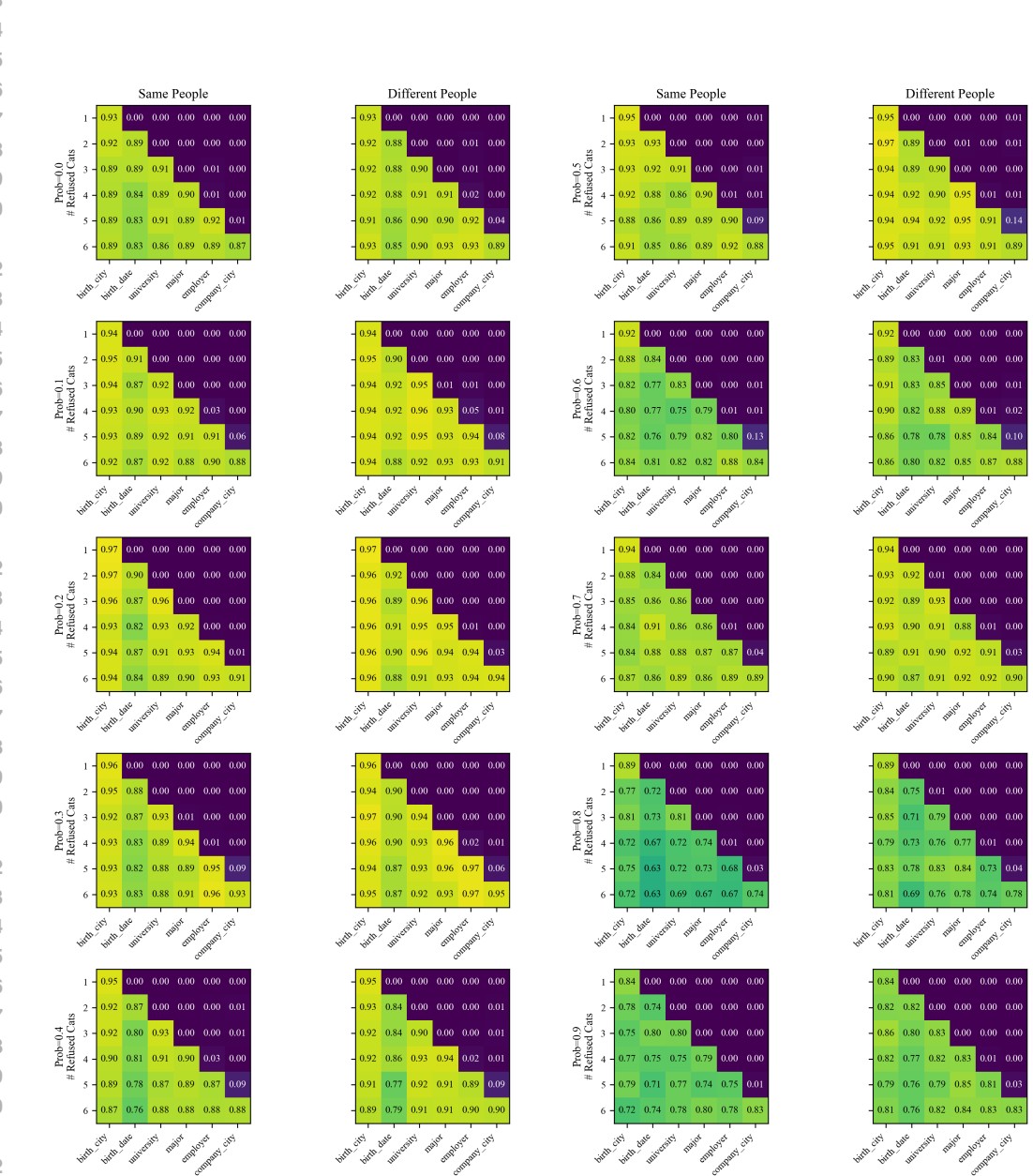

Figure 16: Results of refusal rate with correlation intensity from 0.0 to 0.9. High correlation also causes a decline in refusal rate, and training on some subset of attributes does not contribute to the others. In each heat map, the performance of first row is better than the last row, this is due to our fixed volume data mixing scheme that maintains the refusal data proportion at 12%. More classes share a fixed total amount, this results in a reduced amount of data allocated to each class.

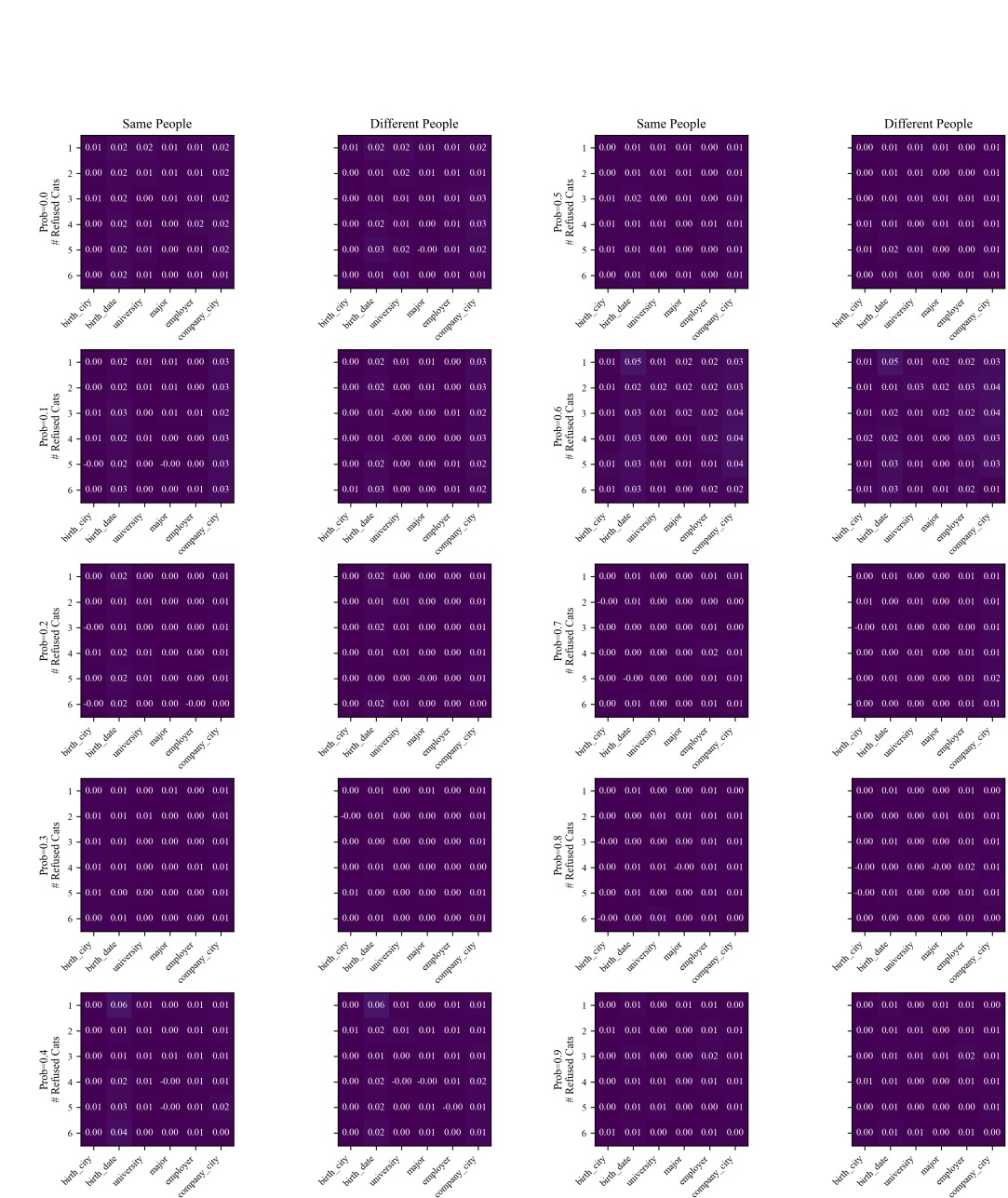

Figure 17: Results of hallucinaion rate, obtained simutaneously while testing accuracy. It shows there is almost no hallucination occurs when evaluating at known individuals. The deteriorated accuracy almost stem from over-refusal.

