# OpenReview forum: "When Bias Pretends to Be Truth: How Spurious Correlations Undermine Hallucination Detection in LLMs"
_ICLR.cc/2026/Conference — Submitted to ICLR 2026_

### Official Review · Reviewer_mVcA · 2025-10-15

**Soundness:** 2
**Presentation:** 2
**Contribution:** 3
**Rating:** 4
**Confidence:** 4

**Summary:**

This paper investigates the severe impact of spurious correlations on hallucination detection methods for large language models (LLMs). The authors construct a controllable synthetic experimental environment by introducing spurious correlations between surnames and attributes, and systematically evaluate the performance of various hallucination detection methods (e.g., confidence-based, internal-state probing, IDK fine-tuning) under different correlation strengths. The experiments show that as spurious correlations increase, the performance of existing detection methods drops significantly, making hallucination detection more difficult; even model scaling and IDK fine-tuning fail to effectively mitigate this issue. Theoretical analysis further reveals that in kernel regression models, spurious correlations cause the model to be overconfident in “rule regions”, making hallucinations harder to detect.

**Strengths:**

1. The problem is important: The paper focuses on spurious correlations, an overlooked source of hallucinations, and exposes the fundamental limitations of existing detection methods, demonstrating strong theoretical and practical significance.

2. Support for reproducibility: Detailed experimental settings, data construction, model configurations, and prompt templates are provided, facilitating reproduction and validation in future research.

**Weaknesses:**

1. Limited experimental scale and model coverage: Although multiple models are included, the experiments mainly focus on small- to medium-scale models (e.g., 1.7B parameters), with insufficient systematic evaluation on larger models (e.g., tens to hundreds of billions of parameters). It is recommended to include results for larger-scale models.

2. Narrow definition of spurious correlations: Currently, spurious correlations are introduced only via surname-attribute mappings, whereas in reality, spurious correlations can take various forms (e.g., context, word frequency, co-occurrence). It is suggested to expand experiments to include more types of spurious correlations.

3. Lack of exploration of effective mitigation strategies: The paper primarily reveals the problem but provides limited discussion on how to mitigate it. Future work could propose and evaluate methods for hallucination mitigation that target spurious correlations.

4. Readability of some figures could be improved: Some figures (e.g., heatmaps, linear probing results) have unclear labels. Optimizing legends and annotations would enhance visualization clarity.

5. Theoretical section is somewhat obscure: The derivations involving kernel regression and NTK may be difficult for readers without a theoretical background. It is recommended to add more intuitive explanations or illustrative diagrams in the main text.

**Questions:**

1. Limited experimental scale and model coverage: Although multiple models are included, the experiments mainly focus on small- to medium-scale models (e.g., 1.7B parameters), with insufficient systematic evaluation on larger models (e.g., tens to hundreds of billions of parameters). It is recommended to include results for larger-scale models.

2. Narrow definition of spurious correlations: Currently, spurious correlations are introduced only via surname-attribute mappings, whereas in reality, spurious correlations can take various forms (e.g., context, word frequency, co-occurrence). It is suggested to expand experiments to include more types of spurious correlations.

3. Lack of exploration of effective mitigation strategies: The paper primarily reveals the problem but provides limited discussion on how to mitigate it. Future work could propose and evaluate methods for hallucination mitigation that target spurious correlations.

4. Readability of some figures could be improved: Some figures (e.g., heatmaps, linear probing results) have unclear labels. Optimizing legends and annotations would enhance visualization clarity.

5. Theoretical section is somewhat obscure: The derivations involving kernel regression and NTK may be difficult for readers without a theoretical background. It is recommended to add more intuitive explanations or illustrative diagrams in the main text.

---

> ### Author Response · Authors · 2025-11-20
>
> Dear Reviewer mVcA:
>
> We would like to first extend our sincere gratitude for your time and effort in evaluating our manuscript. We will address your questions point by point and hope to resolve your concerns effectively.
>
> > Limited experimental scale and model coverage:
>
> We respectfully disagree with this comment. While our main controlled experiments focus on small- to medium-scale models, in Sec. `Validation on Real World LLM`, **we have evaluated four large-scale models, including the 671B-parameter Deepseek-V3 and the commercial LLM GPT5.** These results show that the hallucination caused by spurious correlation we study also appears in larger, real-world LLMs and similarly degrades hallucination detection performance, supporting the generality of our findings beyond the smaller models.
>
> > Narrow definition of spurious correlations
>
> **In the original paper, we introduced two** **types of spurious correlation.** In the synthetic data setting, we use surname-attribute correlation to control the strength of spurious correlation. In real-world LLM validation, we use spurious co-occurrence (strong but non-causal co-occurrence between question entities and entities in wrong answers from the LLM), measured by the Jaccard similarity between question entities and entities in wrong answers, to represent spurious correlation.
>
> **In response to the reviewer's comment, we further introduce a third type of spurious correlation, style correlation**. Here, “style” refers to superficial textual properties (e.g., formality, emotional tone, template structure) that are not causally related to the ground-truth answer. In real data, such correlations naturally arise: mathematical or scientific content is often concise and formal, while literary or social content tends to be more narrative and emotional. These regularities can become shortcuts for the model.
>
> This can induce hallucinations: when the prompt’s style resembles contexts historically associated with a certain answer type, the model may follow this stylistic prior. To isolate this effect, we extend our synthetic data setting by **introducing a controllable correlation between templates and an auxiliary attribute (e.g., profession) at** the **SFT Stage**. Specifically, each profession is more likely to appear with specific templates, although the template is irrelevant to the correct answer. We adopt the exact same experimental settings as the original paper. As shown in the results below, we find that most detection methods' performance drops under stronger format correlation. **Detailed results are included in** Appendix E.2 of the updated paper.
>
> | Methods                      | Without Format Correlation($\rho=0$) | Strong Format Correlation($\rho=0.9$) |
> | ---------------------------- | -------------------------------------- | --------------------------------------- |
> | Linear Probe (Input Last)    | 0.7960                                 | 0.7117                                  |
> | Linear Probe (Output Last)   | 0.7149                                 | 0.7037                                  |
> | Linear Probe (Input Avg)     | 0.7588                                 | 0.6544                                  |
> | Linear Probe (Output Avg)    | 0.6999                                 | 0.6608                                  |
> | Perplexity                   | 0.9961                                 | 0.5863                                  |
> | Window Entropy               | 0.5064                                 | 0.5302                                  |
> | Logit Entropy                | 0.5036                                 | 0.5309                                  |
> | Attention Score (Best Layer) | 0.4854                                 | 0.5108                                  |

---

> ### Author Response · Authors · 2025-11-20
>
> > Lack of exploration of effective mitigation strategies
>
> We believe the primary value of this work lies in empirically uncovering and characterizing a critical failure mode: systematic, high-confidence hallucinations driven by spurious correlations that evade existing detection methods. Notably, **this** **vulnerability** **persists even in advanced commercial models like GPT-5**. Given the magnitude of this challenge, we focus on this diagnosis and leave the design of effective mitigation strategies for future work.
>
> > Readability of some figures could be improved
>
> We have updated our figures to make them more readable.
>
> > Theoretical section is somewhat obscure
>
> We have revised the main text to provide clearer insights into the simplified theoretical model and the hallucination detection criterion, ensuring a smoother transition from experimental findings to theoretical analysis. Specifically, we have expanded the discussion of the relationship between NTK and kernel regression (following Theorem 1, with details in Appendix D) and added clear explanations for Figure 6 to illustrate the impact on fully connected networks.
>
> **The objective of our theoretical section is simple:** we want to prove that the failure of hallucination detection is not caused by the complexity of modern LLMs (such as Transformers), but rather a fundamental result of learning from biased data. **By showing that this failure occurs even in the simplest possible model (a toy** **kernel regression** **model), we demonstrate that spurious correlations alone are enough to break existing confidence-based detection methods.** This confirms that the issue is universal, regardless of how complex the model architecture is.

---

> ### Author Response · Authors · 2025-11-27
>
> Dear Reviewer,
>
> I hope this message finds you well. As the discussion period is nearing its end with less than a week remaining, I wanted to ensure we have addressed all your concerns satisfactorily. If there are any additional points or feedback you'd like us to consider, please let us know. Your insights are invaluable to us, and we're eager to address any remaining issues to improve our work.
>
> Thank you for your time and effort in reviewing our paper.

---

### Official Review · Reviewer_2nau · 2025-10-28

**Soundness:** 3
**Presentation:** 3
**Contribution:** 2
**Rating:** 4
**Confidence:** 2

**Summary:**

The paper identifies an important failure mode of hallucination detection in large language models (LLMs), attributing it to spurious correlations in training data. It provides a rigorous theoretical analysis and carefully designed synthetic and real-world experiments to support the claim that spurious correlation leads to high confident hallucinations  that are hard to detect under current detection schemes.

**Strengths:**

This paper provides rigorous theoretical derivation supporting the their hypothesis. They propose well-controlled synthetic experiments and pair with evolution on real LLMs. The paper demonstrates consistent empirical trend showing the degradation of detection under strong spurious correlations.The paper identifies an important failure mode of hallucination detection in large language models (LLMs), attributing it to spurious correlations in training data. It provides a rigorous theoretical analysis and carefully designed synthetic and real-world experiments to support this claim.

**Weaknesses:**

My main concerns is regarding the novelty and practical applicability of the paper. The core problem — that spurious correlations can lead models to make incorrect predictions — is well established in prior robustness and fairness literature. While this paper contextualizes the issue within LLM hallucination detection and LLM confidence, the theoretical analysis is based on a kernel regression formulation that is too simplified to capture the dynamics of modern LLMs. Moreover, the theoretical bound derived does not appear to provide actionable insights for model design or detection improvement, limiting its practical relevance.

Overall, I find the work interesting and rigorous, but I am uncertain whether the contribution level is sufficient for acceptance as a theory paper.

[1] Yang, Yu, Eric Gan, Gintare Karolina Dziugaite, and Baharan Mirzasoleiman. "Identifying spurious biases early in training through the lens of simplicity bias." In International conference on artificial intelligence and statistics, pp. 2953-2961. PMLR, 2024.

[2] Ye, Wenqian, Guangtao Zheng, Xu Cao, Yunsheng Ma, and Aidong Zhang. "Spurious correlations in machine learning: A survey." arXiv preprint arXiv:2402.12715 (2024).

**Questions:**

see weakness

---

> ### Author Response · Authors · 2025-11-20
>
> Dear Reviewer 2nau:
>
> We would like to first extend our sincere gratitude for your time and effort in evaluating our manuscript. We will address your questions point by point and hope to resolve your concerns effectively.
>
> >  but I am uncertain whether the contribution level is sufficient for acceptance as a theory paper.
>
> We thank the reviewer for the thoughtful comments regarding the positioning of our work. We wish to clarify that **our primary contribution is empirical**, and we do not intend to frame this as a purely theoretical paper. The theoretical analysis serves as a **simplified modeling framework** designed to rigorously explain the failure mechanisms of hallucination detection methods observed in our experiments. To better reflect this priority, we have restructured the revised paper so that the controlled synthetic and real-world LLM experiments now appear before the theoretical analysis.
>
> > The core problem — that spurious correlations can lead models to make incorrect predictions — is well established in prior robustness and fairness literature.
>
> While it is widely recognized that spurious correlations can lead to model errors, our research emphasizes a specific and severe consequence within the context of large language models (LLMs): **systematic, high-confidence hallucinations that current hallucination detection/mitigation methods consistently fail to identify or mitigate.** Even in the absence of spurious correlations, factors such as limited model size[1][2] or inappropriate post-training[3][4] can still induce hallucinations in LLMs. A core research question receiving significant attention within the LLM safety community involves effectively detecting[5][6] or mitigating these hallucinations[7][8]. To our knowledge, the influence of spurious correlations on the reliability of existing hallucination detection and mitigation strategies in LLMs has received little systematic attention.
>
> > the theoretical analysis is based on a kernel regression formulation that is too simplified to capture the dynamics of modern LLMs.
>
> We employ this simplified theoretical model to demonstrate that the impact of spurious correlations is a fundamental phenomenon that **emerges even in a minimal setting**. Our goal is to show that the failure modes observed in complex LLMs are not unique to their architecture, but can be reproduced and analyzed within a basic kernel ridgeless regression framework. This indicates that our abstraction successfully captures the **core mechanism**—the interplay between data interpolation and spurious features—without the confounding complexity of modern LLMs. As clarified in Appendix D, kernel regression serves as a valid surrogate given its connection to the interpolation behavior of neural networks (via NTK/NNGP limits).
>
> > practical applicability of the paper.
>
> We believe **this failure mode is practically important**: it appears across multiple detectors, in both carefully controlled synthetic setups and real-world tasks, and persists even for state-of-the-art LLMs (e.g., GPT 5). We hope our work, by identifying a critical blind spot in current LLM safety practices, can **serve as a testbed for the community to evaluate future detection and mitigation methods** and improve LLM robustness.
>
>
> [1]Obaid Ul Islam, Saad et al. “How Much Do LLMs Hallucinate across Languages? On Multilingual Estimation of LLM Hallucination in the Wild.” ArXiv abs/2502.12769 (2025): n. pag.
>
> [2]Pan, Zhixuan et al. “Understanding LLM Behaviors via Compression: Data Generation, Knowledge Acquisition and Scaling Laws.” ArXiv abs/2504.09597 (2025): n. pag.
>
> [3]Gekhman, Zorik et al. “Does Fine-Tuning LLMs on New Knowledge Encourage Hallucinations?” ArXiv abs/2405.05904 (2024): n. pag.
>
> [4]Song, Linxin et al. “The Hallucination Tax of Reinforcement Finetuning.” ArXiv abs/2505.13988 (2025): n. pag.
>
> [5]Manakul, Potsawee et al. “SelfCheckGPT: Zero-Resource Black-Box Hallucination Detection for Generative Large Language Models.” ArXiv abs/2303.08896 (2023): n. pag.
>
> [6]Sriramanan, Gaurang et al. “LLM-Check: Investigating Detection of Hallucinations in Large Language Models.” Advances in Neural Information Processing Systems 37 (2024): n. pag.
>
> [7]Dhuliawala, Shehzaad et al. “Chain-of-Verification Reduces Hallucination in Large Language Models.” Annual Meeting of the Association for Computational Linguistics (2023).
>
> [8]Zhang, Hanning et al. “R-Tuning: Teaching Large Language Models to Refuse Unknown Questions.” ArXiv abs/2311.09677 (2023): n. pag.

---

> ### Author Response · Authors · 2025-11-27
>
> Dear Reviewer,
>
> I hope this message finds you well. As the discussion period is nearing its end with less than a week remaining, I wanted to ensure we have addressed all your concerns satisfactorily. If there are any additional points or feedback you'd like us to consider, please let us know. Your insights are invaluable to us, and we're eager to address any remaining issues to improve our work.
>
> Thank you for your time and effort in reviewing our paper.

---

### Official Review · Reviewer_5ieG · 2025-11-02

**Soundness:** 3
**Presentation:** 4
**Contribution:** 3
**Rating:** 8
**Confidence:** 3

**Summary:**

The paper studies the cause of a specific kind of hallucinations, when spurious correlations in the data cause LLMs to confidently generate hallucinated content. Through controlled synthetic data usage for pretraining and fine-tuning, the paper shows how spurious correlations present in the data can make the LLM learn 'shortcut' connections, resulting in hallucinations that are not detected by most SOTA detection techniques, and not mitigated by most SOTA mitigation techniques.

**Strengths:**

1. The paper focuses on a targeted research question, and does a good job of providing an in-depth discussion.
2. The theoretical discussion and the controlled experiments with synthetic data across a wide range of models both strongly support the hypothesis proposed in the paper.
3. The paper is well written, and I enjoyed reading it. The Related Work is mostly well done, the Theoretical discussion is broadly easy to follow (although it can be made more accessible), and the experiment details are clear based on the main paper, the appendix, and the supplementary code (I strongly recommend adding a README to the code, though).

**Weaknesses:**

1. My only complaint, the 'real world validation' is not as strong, and thus leaves open the question of how to actually recognize spurious correlations in the wild. It is not clear to me how the Jaccard similarity of entities between the question and the generated answer is a measure of spurious correlations. While I do believe this is a good paper, even without the real world validation, clarification of these experiments would be appreciated. In their current form, I'm not convinced these results support the larger story.

**Questions:**

1. Why is the Jaccard similarity of entities between the question and the generated answer a measure of spurious correlations?

A Comment on Related Works
------
There has been a lot of work recently on trying to map out the 'knowledge' of an LLM [1, 2], or 'consistency' of hallucination evaluations [3]. While not exactly the same, these related works can also provide an interesting discussion for the paper. In my opinion, incorrect 'knowledge' in LLMs (or 'consistent' hallucinations) is probably an effect of spurious correlations detected in this paper.

References -

[1] Yin, Xunjian, et al. "Benchmarking Knowledge Boundary for Large Language Models: A Different Perspective on Model Evaluation." Proceedings of the 62nd Annual Meeting of the Association for Computational Linguistics (Volume 1: Long Papers). 2024.

[2] Gekhman, Zorik, et al. "Inside-out: Hidden factual knowledge in llms." arXiv preprint arXiv:2503.15299 (2025).

[3] Ganesh, Prakhar, et al. "Rethinking hallucinations: Correctness, consistency, and prompt multiplicity." ICLR 2025 Workshop on Building Trust in Language Models and Applications. 2025.

---

> ### Author Response · Authors · 2025-11-20
>
> Dear Reviewer 5ieG:
>
> We would like to first extend our sincere gratitude for your time and effort in evaluating our manuscript. We will address your question and hope to resolve your concerns effectively.
>
> > Why is the Jaccard similarity of entities between the question and the generated answer a measure of spurious correlations?
>
> We believe that spurious co-occurrence (strong but non-causal co-occurrence with question entities), as measured by Jaccard similarity of entities of question and entities of wrong answer from the LLM, is one special form of spurious correlation. We show an example below:
>
> ```
> Q: To which academic society was computer scientist Sarita Vikram Adve elected in 2020?
> Answer from GPT5:  Association for Computing Machinery
> Ground Truth Answer: American Academy of Arts and Sciences
> ```
>
> Table 1: Jaccard Similarity between entities from the question and entities from answers
> | | computer scientist | Sarita Vikram Adve |
> | :--- | :--- | :--- |
> | Association for Computing Machinery | **0.0785** | 0.0006 |
> | American Academy of Arts and Sciences | 0.0099 | 0.0002 |
>
> As shown in the table above,  "Association for Computing Machinery" often correlates with "computer scientist". As a result, even if the Wikipedia dataset includes the information that Adve was elected to the American Academy of Arts and Sciences (https://en.wikipedia.org/wiki/Sarita_Adve), the model still hallucinates due to spurious correlation.
>
> In the revised manuscript, we have included this example in Appendix E.1 to clarify our argument. Thank you again for your encouraging review.

---

> ### Author Response · Authors · 2025-11-27
>
> Dear Reviewer,
>
> I hope this message finds you well. As the discussion period is nearing its end with less than a week remaining, I wanted to ensure we have addressed all your concerns satisfactorily. If there are any additional points or feedback you'd like us to consider, please let us know. Your insights are invaluable to us, and we're eager to address any remaining issues to improve our work.
>
> Thank you for your time and effort in reviewing our paper.

---

> > ### Comment · Reviewer_5ieG · 2025-11-27
> >
> > The authors' response is appreciated. The reasoning behind using Jaccard similarity as a measure of spurious correlations still feels weak to me. This is only one special kind of spurious correlation where the generated answer simply reuses terms from the question. Still, I continue to be on the positive side for this paper.

---

> > > ### Author Response · Authors · 2025-12-04
> > >
> > > We agree with the reviewer that the Jaccard similarity we described is indeed a form of spurious correlation. However, we highlight that this observation reveals a critical vulnerability: even such a specific spurious pattern is sufficient to significantly mislead SOTA LLMs. To further demonstrate the generality of this issue, we have added new experiments on style-related spurious correlations in the updated paper, confirming that models remain highly susceptible to various forms of these non-robust features and are harder to detect or mitigate.

---

### Author Response · Authors · 2025-12-04

We thank the Area Chair for their time and supervision. Below, we provide a brief summary of the paper, the status of reviewer discussions, and the new experimental results added during the rebuttal to address concerns regarding the generality of spurious correlations.

**What is "When Bias Pretends to Be Truth"?**

This paper identifies a critical safety vulnerability: **spurious correlations** (both semantic and stylistic) in training data induce high-confidence hallucinations that are **immune to model scaling and evade state-of-the-art detection and mitigation methods** (such as confidence-based filtering and inner-state probing). We demonstrate this failure mode through rigorous theoretical analysis, controlled synthetic experiments, and validation on real-world LLMs (including GPT-5). We call for future research to explore novel approaches specifically targeting the identification and mitigation of these problematic correlations throughout the model development lifecycle

**Strengths Highlighted Across Reviews**

All reviewers acknowledged the significance of the problem and the rigorousness of the approach:
* **Targeted & Important Problem:** The paper addresses an "overlooked source of hallucinations" (mVcA) and provides an "in-depth discussion" (5ieG) on why current detection fails.
* **Theoretical Rigor:** The theoretical derivation is "rigorous" and "strongly support[s] the hypothesis" (5ieG, 2nau).
* **Reproducibility:** Detailed experimental settings and data construction facilitate future research (mVcA).
* **Presentation:** The paper is well-written, with clear experimental details (5ieG).

**Reviewer Discussion After Rebuttal**
* **Reviewer 5ieG (Score 8):** Strongly positive. While they initially questioned the Jaccard similarity metric, they acknowledged our response and remained positive ("I continue to be on the positive side"). To further address their comment on generality, we added new experiments on **style/format correlations** (detailed below).
* **Reviewer 2nau (Score 4):** Raised concerns about the simplicity of the theoretical model. In our rebuttal, we clarified that the theory serves as a foundational abstraction to explain the empirical failure modes observed in complex models, and we restructured the paper to prioritize empirical findings, **as our main contribution is empirical**. The reviewer did not respond during the discussion period due to the ICLR's new policy.
* **Reviewer mVcA (Score 4):** Raised concerns about model scale and the definition of spurious correlation. We clarified that **our evaluation already includes 671B (DeepSeek-V3) and GPT-5**, proving scalability. We also expanded the definition of spurious correlations to include "style/format" shortcuts. The reviewer did not respond during the discussion period due to the ICLR's new policy.

**New Experiments: Style & Format Correlations**

To address concerns from **Reviewer 5ieG** (regarding Jaccard similarity generality) and **Reviewer mVcA** (regarding the narrow definition of spurious correlations), we introduced a new set of experiments in **Appendix E.2**.

We simulated **style-based spurious correlations** by creating strong associations between specific prompt templates (formats) and target attributes. The results confirm that our findings generalize beyond simple surname-attribute associations: as format correlation increases ($\rho=0.9$), detection performance collapses across all metrics.

| Methods | Without Format Correlation ($\rho=0$) | Strong Format Correlation ($\rho=0.9$) |
| :--- | :--- | :--- |
| **Linear Probe (Input Last)** | 0.7960 | **0.7117** |
| **Linear Probe (Output Last)** | 0.7149 | **0.7037** |
| **Linear Probe (Input Avg)** | 0.7588 | **0.6544** |
| **Linear Probe (Output Avg)** | 0.6999 | **0.6608** |
| **Perplexity** | 0.9961 | **0.5863** |
| **Window Entropy** | **0.5064** | 0.5302 |
| **Logit Entropy** | **0.5036** | 0.5309 |

These results reinforce the core claim of the paper: effectively detecting hallucinations driven by *any* form of strong spurious correlation (semantic or stylistic) remains an unsolved challenge for current safety techniques.

---

### Meta-Review · Area_Chair_EYtN · 2026-01-11

**Summary:**

This work identifies and systematically studies a previously underexplored source of hallucinations in LLMs arising from spurious correlations in the training data. Through controlled synthetic experiments and evaluations on both open-source and proprietary models, the authors show that such hallucinations are confidently produced, persist under scaling and refusal fine-tuning, and evade existing detection methods, including confidence-based filtering and internal probing. The paper further provides a theoretical analysis explaining why spurious correlations fundamentally undermine confidence-based hallucination detection.

During the rebuttal, the authors successfully addressed some of the reviewers’ concerns, but several major issues remain. First, it is widely accepted in the robustness community that spurious correlations mainly arise from biased data, which is also demonstrated by the authors’ theoretical analysis and highlighted in their rebuttal: “we want to prove that the failure of hallucination detection is not caused by the complexity of modern LLMs (such as Transformers), but rather a fundamental result of learning from biased data.” If hallucination detection is viewed as analogous to out-of-distribution (OOD) detection in neural networks, then the main finding of this work is not particularly surprising.

While the authors provide a comprehensive analysis of hallucination detection in LLMs, we encourage them to take Reviewer mVcA’s feedback and propose a corresponding mitigation method motivated by the insights of this study, in order to strengthen the contribution. Taken together, we recommend rejecting this work in its current form.

**Reviewer Concerns:**

Reviewer 5ieG’s (score: 8) concerns have been well addressed. The authors provided a clearer justification for using Jaccard similarity as a measure of spurious correlations.

Reviewer 2nau’s (score: 4) concern that spurious correlations causing incorrect predictions are already well established in the robustness and fairness literature remains.

Reviewer mVcA’s (score: 4) concern regarding the lack of exploration of effective mitigation strategies also remains unresolved.

**Reviewer Scores:**

Since the major concerns of Reviewer 2nau (score: 4)  and Reviewer mVcA (score: 4) did not get addressed, there is a low chance that they would change their score.

Reviewer 5ieG (score: 8)  would highly maintain the original positive score.

---

### Decision · Program_Chairs · 2026-01-26

Reject